# SEMANTIC ENTROPY PROBES: ROBUST AND CHEAP HALLUCINATION DETECTION IN LLMS

## ABSTRACT

We propose semantic entropy probes (SEPs), a cheap and reliable method for uncertainty quantification in Large Language Models (LLMs). Hallucinations, which are plausible-sounding but factually incorrect and arbitrary model generations, present a major challenge to the practical adoption of LLMs. Recent work by Farquhar et al. (2024) proposes semantic entropy (SE), which can reliably detect hallucinations by quantifying the uncertainty over different generations by estimating entropy over semantically equivalent sets of outputs. However, the 5-to-10-fold increase in computation cost associated with SE computation hinders practical adoption. To address this, we propose SEPs, which directly approximate SE from the hidden states of a single generation. SEPs are simple to train and do not require sampling multiple model generations at test time, reducing the overhead of semantic uncertainty quantification to almost zero. We show that SEPs retain high performance for hallucination detection and generalize better to out-of-distribution data than previous probing methods that directly predict model accuracy. Our results across models and tasks suggest that model hidden states capture SE, and our ablation studies give further insights into the token positions and model layers for which this is the case.

## 1 INTRODUCTION

Large Language Models (LLMs) have demonstrated impressive capabilities across a wide variety of natural language processing tasks (Touvron et al., 2023a;b; OpenAI, 2023; Team, 2023; Brown et al., 2020). They are increasingly deployed in real-world settings, including in high-stakes domains such as medicine, journalism, or legal services (Singhal et al., 2023; Weiser, 2023; Opdahl et al., 2023; Shen et al., 2023). It is therefore paramount that we can *trust* the outputs of LLMs. Unfortunately, LLMs have a tendency to *hallucinate*. Originally defined as "content that is nonsensical or unfaithful to the provided source" (Maynez et al., 2020; Filippova, 2020; Ji et al., 2023), the term is now used to refer to nonfactual, arbitrary content generated by LLMs. For example, when asked to generate biographies, even capable LLMs such as GPT-4 will often fabricate facts entirely (Min et al., 2023; Tian et al., 2024; Farquhar et al., 2024). While this may be acceptable in low-stakes use cases, hallucinations can cause significant harm when factuality is critical. The reliable detection or mitigation of hallucinations is a key challenge to ensure the safe deployment of LLM-based systems.

Various approaches have been proposed to address hallucinations in LLMs (see Section 2). An effective strategy for detecting hallucinations is to sample multiple responses for a given prompt and check if the different samples convey the same meaning (Farquhar et al., 2024; Kuhn et al., 2023; Kadavath et al., 2022; Duan et al., 2023; Cole et al., 2023; Chen & Mueller, 2023; Elaraby et al., 2023; Manakul et al., 2023b; Min et al., 2023). The core idea is that if the model knows the answer, it will consistently provide the same answer. If the model is hallucinating, its responses may vary across generations. For example, given the prompt "What is the capital of France?", an LLM that "knows" the answer will consistently output (Paris, Paris, Paris), while an LLM that "does not know" the answer may output (Naples, Rome, Berlin), indicating a hallucination.

One explanation for why this works is that LLMs have calibrated uncertainty (Kadavath et al., 2022; OpenAI, 2023), i.e., "language models (mostly) know what they know" (Kadavath et al., 2022). When an LLM is certain about an answer, it consistently provides the correct response. Conversely, when uncertain, it generates arbitrary answers. This suggests that we can leverage model uncertainty to detect hallucinations. However, we cannot use token-level probabilities to estimate uncertainty

directly because different sequences of tokens may convey the same meaning. For the example, the answers "Paris", "It's Paris", and "The capital of France is Paris" all mean the same. To address this, Farquhar et al. (2024) propose *semantic entropy* (SE), which clusters generations into sets of equivalent meaning and then estimates uncertainty in semantic space.

A major limitation of SE and other sampling-based approaches is that they require multiple model generations for each input query, typically between 5 and 10. This results in a 5-to-10-fold higher cost compared to naive generation without SE, presenting a major hurdle to the practical adoption of these methods. Computationally cheaper methods for reliable hallucination detection in LLMs are needed.

The hidden states of LLMs are a promising avenue to better understand, predict, and steer a wide range of LLM behaviors (Zou et al., 2023; Hernandez et al., 2023; Subramani et al., 2022). In particular, a recent line of work learns to predict the truthfulness of model responses by training a simple linear probe on the hidden states of LLMs. Linear probes are computationally efficient, both to train and when used at inference. However, existing approaches are usually supervised (Rimsky et al., 2023; Li et al., 2024; Azaria & Mitchell, 2023;

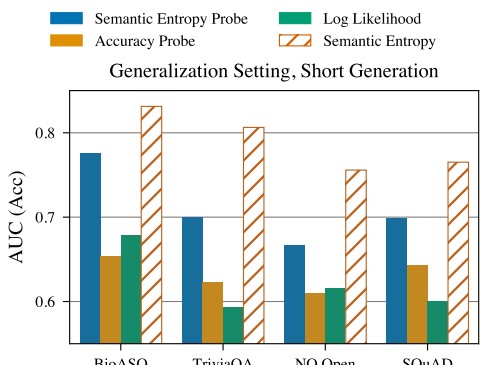

Figure 1: Semantic entropy probes (SEPs) outperform cheap hallucination detection methods, such as accuracy probes and log likelihood, but fall short compared to the much costlier and slower SE baseline for Llama-2-7B. See Sec. 5.

Marks & Tegmark, 2023) and therefore require a labeled training dataset assigning accuracy to statements or model generations. And while unsupervised approaches exist (Burns et al., 2023), their validity has been questioned (Farquhar et al., 2023). In this paper, we argue that supervising probes via *SE* is preferable to accuracy labels for robust prediction of truthfulness.

We propose *Semantic Entropy Probes* (SEPs), linear probes that capture semantic uncertainty from the hidden states of LLMs, presenting a cost-effective and reliable hallucination detection method. SEPs combine the advantages of probing and sampling-based hallucination detection. Like other probing approaches, SEPs are easy to train, cheap to deploy, and can be applied to the hidden states of a single model generation. Similar to sampling-based hallucination detection, SEPs capture the *semantic uncertainty* of the model. Furthermore, they address some of the shortcomings of previous approaches. Contrary to sampling-based hallucination detection, SEPs act directly on a *single* model hidden state and do not require generating multiple samples at test time. And unlike previous probing methods, SEPs are trained to predict *semantic entropy* (Farquhar et al., 2024) rather than model accuracy, which can be computed without access to ground truth accuracy labels that can be expensive to curate.

We find that SEP predictions are effective proxies for truthfulness. In fact, SEPs generalize better to new tasks than probes trained directly to predict accuracy, setting a new state-of-the-art for cost-efficient hallucination detection, cf. Fig. 1. Our results additionally provides insights into the inner workings of LLMs, strongly suggesting that model hidden states directly capture the model's uncertainty over semantic meanings. Through ablation studies, we show that this holds across a variety of models, tasks, layers, and token positions.

In summary, our core contributions are:

- We propose Semantic Entropy Probes (SEPs), linear probes trained on the hidden states of LLMs to capture semantic entropy (Section 4).

- We demonstrate that semantic entropy is encoded in the hidden states of a single model generation and can be successfully extracted using probes (Section 6).

- We perform ablation studies to study SEP performance across models, tasks, layers, and token positions. Our results strongly suggest internal model states across layers and tokens implicitly capture semantic uncertainty, even before generating any tokens. (Section 6)

- We show that SEPs can be used to predict hallucinations and that they generalize better than probes directly trained for accuracy as suggested by previous work, establishing a new state-of-the-art for cost-efficient hallucination detection (Section 7, Fig. 1).

## 2 RELATED WORK

**LLM Hallucinations.** We refer to Rawte et al. (2023); Zhang et al. (2023b) for surveys on hallucinations in LLMs and here review the most relevant related work to this paper. Early work on hallucinations in language models typically refers to issues in summarization tasks where models "hallucinate" content that is not faithful to the provided source text (Maynez et al., 2020; Deutsch et al., 2021; Durmus et al., 2020; Cao et al., 2022; Wang et al., 2020; Manakul et al., 2023a; Nan et al., 2021). Around the same time, research emerged that showed LLMs themselves could store and retrieve factual knowledge (Petroni et al., 2019), leading to the currently popular closed-book setting, where LLMs are queried without any additional context (Roberts et al., 2020). Since then, a large variety of work has focused on mitigating hallucinations in LLMs, and we next give an overview to popular approaches.

**Sampling-Based Hallucination Detection.** A variety of methods have been proposed that sample multiple model completions for a given query and then quantify the semantic difference between the model generations (Kuhn et al., 2023; Kadavath et al., 2022; Duan et al., 2023; Cole et al., 2023; Chen & Mueller, 2023; Elaraby et al., 2023). For this paper, Farquhar et al. (2024) is particularly relevant, as we use their semantic entropy measure to supervise our hidden state probes, cf. Section 3. A different line of work does not directly re-sample answers for the same query, but instead asks follow-up questions to uncover inconsistencies in the original answer (Dhuliawala et al., 2023; Agrawal et al., 2024). Recent work has also extended hallucination detection to scenarios where models generate longer paragraphs of text by decomposing generations into individual facts or sentences, and then validating those facts separately (Luo et al., 2023; Mündler et al., 2023; Manakul et al., 2023b; Dhuliawala et al., 2023).

**Retrieval-Based Methods.** A different strategy is to rely on external knowledge bases, e.g. web search, to verify the factuality of model responses (Feldman et al., 2023; Zhang et al., 2023a; Peng et al., 2023; Dziri et al., 2021; Gao et al., 2022; Li et al., 2023; Varshney et al., 2023; Su et al., 2022). Such methods need not rely on good model uncertainties and can be used directly to fix errors in model generations. However, retrieval-based approaches can add significant cost and latency. Further, they may be less effective for domains such as reasoning, where LLMs are also prone to produce unfaithful and misleading generations (Turpin et al., 2023; Lanham et al., 2023). Thus, retrieval- and uncertainty-based methods are orthogonal and can be combined for maximum effect.

**Sampling and Finetuning Strategies.** Prior work has also proposed to reduce hallucinations in LLMs through sampling schemes (Lee et al., 2022; Chuang et al., 2024; Shi et al., 2023), preference optimization targeting factuality (Tian et al., 2024), or finetuning to align "verbal" uncertainties of LLMs with model accuracy (Mielke et al., 2022; Lin et al., 2023; Band et al., 2024).

**Hidden State Methods.** Azaria & Mitchell (2023) show that the factuality of LLM generations can be predicted from hidden state probes, which are trained from a set of model generations and associated accuracy labels. The success of accuracy probes has been replicated by Liu et al. (2024); Ji et al. (2024), with the latter additionally showing that latent states can be used to predict training set membership of queries. In this paper, we follow this line of work and implement a simple linear probe, supervised by model accuracy signals, as one of our baseline methods. Recently, He et al. (2024) have extended accuracy-probing methods to predict factuality on a word-level. While attempts have been made to propose unsupervised methods for factuality probing, these methods fall short in a variety of ways: Burns et al. (2023) is limited to binary questions and Farquhar et al. (2023) further question core assumptions of the approach; Zou et al. (2023) propose a mostly unsupervised prompting strategy but require accuracy labels to select hyperparameters; Chen et al. (2024) calculate a measure of the semantic consistency of latent states, which requires sampling multiple expensive model generations.

## 3 SEMANTIC ENTROPY

Measuring uncertainty in free-form natural language generation tasks is challenging. The uncertainties over tokens output by the language model can be misleading because they conflate semantic uncertainty, uncertainty over the meaning of the generation, with lexical and syntactic uncertainty, uncertainty over how to phrase the answer (see the example in Section 1). To address this, Farquhar et al. (2024); Kuhn et al. (2023) propose *semantic entropy*, which aggregates token-level uncertainties across clusters of semantic equivalence. Semantic entropy is important in the context of this paper because we use it as the supervisory signal to train our hidden state SE probes.

Semantic entropy is calculated in three steps: (1) for a given query $x$, sample model completions from the LLM, (2) aggregate the generations into clusters $(C_1, \ldots, C_K)$ of equivalent semantic meaning, (3) calculate semantic entropy, $H_{\text{SE}}$, by aggregating uncertainties within each cluster. Step (1) is trivial, and we detail steps (2) and (3) below.

**Semantic Clustering.** To determine if two generations convey the same meaning, Farquhar et al. (2024) use natural language inference (NLI) models, such as DeBERTa (He et al., 2021), to predict entailment between the generations. Concretely, two generations $s_a$ and $s_b$ are identical in meaning if $s_a$ entails $s_b$ and $s_b$ entails $s_a$, i.e. they entail each other bi-directionally. Farquhar et al. (2024) then propose a greedy algorithm to cluster generations semantically: for each sample $s_a$, we either add it to an existing cluster $C_k$ if bi-directional entailment holds between $s_a$ and a sample $s_b \in C_k$, or add it to a new cluster if the semantic meaning of $s_a$ is distinct from all existing clusters. After processing all generations, we obtain a clustering of the generations into $K$ distinct semantic meanings.

**Semantic Entropy.** Given an input context $x$, the joint probability of a generation $s$ consisting of tokens $(t_1, \ldots, t_n)$ is given by the product of conditional token probabilities in the sequence, $p(s \mid x) = \prod_{i=1}^n p(t_i \mid t_{1:i-1}, x)$. The probability of the semantic cluster $C$ is then the aggregate probability of all possible generations $s$ which belong to that cluster, $p(C \mid x) = \sum_{s \in C} p(s \mid x)$. The uncertainty associated with the distribution over semantic clusters is the semantic entropy,

$$H[C \mid x] = \mathbb{E}_{p(C|x)}[-\log p(C \mid x)].$$

**Estimating SE in Practice.** In practice, we cannot compute the above exactly. The expectations with respect to $p(s|x)$ and $p(C|x)$ are intractable, as the number of possible token sequences grows exponentially with sequence length. Instead, Farquhar et al. (2024) sample $N$ generations $(s_1, \ldots, s_N)$ at non-zero temperature from the LLM (typically and also in this paper $N = 10$). They then treat $(C_1, \ldots, C_K)$ as Monte Carlo samples from the true distribution over semantic clusters $p(C|x)$, and approximate semantic entropy as

$$H[C \mid x] \approx -1/K \sum_{k=1}^K \log p(C_k|x). \tag{1}$$

We here use an additional approximation, employing the *discrete* variant of SE that yields good performance without access to token probabilities, making it compatible with black-box models (Farquhar et al., 2024). For the discrete SE variant, we estimate cluster probabilities $p(C|x)$ as the fraction of generations in that cluster, $p(C_k|x) = \sum_{j=1}^N \mathbb{1}[s_j \in C_k]/K$, and then compute semantic entropy as the entropy of the resulting categorical distribution, $H_{\text{SE}}(x) := -\sum_{k=1}^K p(C_k|x) \log p(C_k|x)$. Discrete SE further avoids problems when estimating Eq. (1) for generations of different lengths (Malinin & Gales, 2021; Murray & Chiang, 2018; Kuhn et al., 2023; Farquhar et al., 2024).

## 4 SEMANTIC ENTROPY PROBES

Although semantic entropy is effective at detecting hallucinations, its high computational cost may limit its use to only the most critical scenarios. In this section, we propose **Semantic Entropy Probes** (SEPs), a novel method for cost-efficient and reliable uncertainty quantification in LLMs. SEPs are linear probes trained on the hidden states of LLMs to capture semantic entropy (Farquhar et al., 2024). However, unlike semantic entropy and other sampling-based approaches, SEPs act on the hidden states of a *single* model generation and do not require sampling multiple responses from the model at test time. Thus, SEPs solve a key practical issue of semantic uncertainty quantification by almost completely eliminating the computational overhead of semantic uncertainty estimation at test time. We further argue that SEPs are advantageous to probes trained to directly predict model accuracy. Our intuition for this is that semantic entropy is an inherent property of the model that should be encoded in the hidden states and thus should be easier to extract than truthfulness, which relies on potentially noisy external information. We discuss this further in Section 8.

**Training SEPs.** SEPs are constructed as linear logistic regression models, trained on the hidden states of LLMs to predict semantic entropy. We create a dataset of $(h_p^l(x), H_{\text{SE}}(x))$ pairs, where $x$ is an input query, $h_p^l(x) \in \mathbb{R}^d$ is the model hidden state at token position $p$ and layer $l$, $d$ is the hidden state dimension, and $H_{\text{SE}}(x) \in \mathbb{R}$ is the semantic entropy. That is, given an input query $x$, we first generate a high-likelihood model response via greedy sampling and store the hidden state at a particular layer and token position, $h_p^l(x)$. We then sample $N = 10$ responses from the model at high temperature ($T = 1$) and compute semantic entropy, $H_{\text{SE}}(x)$, as detailed in the previous section.

For inputs, we rely on questions from popular QA datasets (see Section 5 for details), although we do not need the ground-truth labels provided by these datasets and could alternatively compute semantic entropy for any unlabeled set of suitable LLM inputs.

**Binarization.** Semantic entropy scores are real numbers. However, for the purposes of this paper, we convert them into binary labels, indicating whether semantic entropy is high or low, and then train a logistic regression classifier to predict these labels. Our motivation for doing so is two-fold. For one, we ultimately want to use our probes for predicting binary model correctness, so we eventually need to construct a binary classifier regardless. Additionally, we would like to compare the performance of SE probes and accuracy probes. This is easier if both probes target binary classification problems. We note that the logistic regression classifier returns probabilities, such that we can always recover fine-grained signals even after transforming the problem into binary classification.

More formally, we compute $\tilde{H}_{\mathrm{SE}}(x) = \mathbb{1}[H_{\mathrm{SE}}(x) > \gamma^\star]$, where $\gamma^\star$ is a threshold that optimally partitions the raw SE scores into high and low values according to the following objective:

$$\gamma^\star = \arg\min_\gamma \sum_{j \in \mathrm{SE}_{\mathrm{low}}} (H_{\mathrm{SE}}(x_j) - \hat{H}_{\mathrm{low}})^2 + \sum_{j \in \mathrm{SE}_{\mathrm{high}}} (H_{\mathrm{SE}}(x_j) - \hat{H}_{\mathrm{high}})^2, \quad (2)$$

where

$$\mathrm{SE}_{\mathrm{low}} = \{j : H_{\mathrm{SE}}(x_j) < \gamma\}, \qquad\qquad \mathrm{SE}_{\mathrm{high}} = \{j : H_{\mathrm{SE}}(x_j) \geq \gamma\},$$

$$\hat{H}_{\mathrm{low}} = \frac{1}{|\mathrm{SE}_{\mathrm{low}}|} \sum_{j \in \mathrm{SE}_{\mathrm{low}}} H_{\mathrm{SE}}(x_j), \qquad \hat{H}_{\mathrm{high}} = \frac{1}{|\mathrm{SE}_{\mathrm{high}}|} \sum_{j \in \mathrm{SE}_{\mathrm{high}}} H_{\mathrm{SE}}(x_j).$$

This procedure is inspired by splitting objectives used in regression trees (Loh, 2011) and we have found it to perform well in practice compared to alternatives such as soft labelling, cf. Appendix B.

In summary, given a input dataset of queries, $\{x_j\}_{j=1}^Q$, we compute a training set of hidden state – binarized semantic entropy pairs, $\{(h_p^l(x_j), \tilde{H}_{\mathrm{SE}}(x_j))\}_{j=1}^Q$, and use this to train a linear classifier, which is our semantic entropy probe (SEP). At test time, SEPs predict the probability that a model generation for a given input query $x$ has high semantic entropy.

**Probing Locations.** We collect hidden states, $h_p^l(x)$, across all layers, $l$, of the LLM to investigate which layers best capture semantic entropy. We consider two different token positions, $p$. Firstly, we consider the hidden state at the last token of the *input* $x$, i.e. the token before generating (TBG) the model response. Secondly, we consider the last token of the *model response*, which is the token before the end-of-sequence token, i.e. the second last token (SLT). We refer to these scenarios as TBG and SLT. The TBG experiments allow us to study to what extent LLM hidden states capture semantic entropy *before* generating a response. The TBG setup potentially allows us to quantify the semantic uncertainty given an input in a single forward pass – without generating any novel tokens – further reducing the cost of our approach over sampling-based alternatives. In practice, this may be useful to quickly determine if a model will answer a particular input query with high certainty.

## 5 EXPERIMENT SETUP

We investigate and evaluate Semantic Entropy Probes (SEPs) across a range of models and datasets. First, we show that we can accurately predict semantic entropy from the hidden states of LLMs (Section 6). We then explore how SEP predictions vary across different tasks, models, tokens indices, and layers. Second, we demonstrate that SEPs are a cheap and reliable method for hallucination detection (Section 7), which generalizes better to novel tasks than accuracy probes, although they cannot match the performance of much more expensive sampling-based methods in our experiments.

**Tasks.** We evaluate SEPs on four datasets: TriviaQA (Joshi et al., 2017), SQuAD (Rajpurkar et al., 2018), BioASQ (Tsatsaronis et al., 2015), and NQ Open (Kwiatkowski et al., 2019). We use the input queries of these tasks to derive training sets for SEPs and evaluate the performance of each method on the validation/test sets, creating splits if needed. We consider a short- and a long-form setting: Short-form answers are generated by few-shot prompting the LLM to answer "as briefly as possible" and long-form answers are generated by prompting for a "single brief but complete sentence", leading to an approximately six-fold increase in the number of generated tokens (Farquhar et al., 2024). Following Farquhar et al. (2024), we assess model accuracy via the SQuAD F1 score for short-form generations, and we use use GPT-4 (OpenAI, 2023) to compare model answers to ground truth labels for long-form answers. We provide prompt templates in Appendix B.2.

**Models.** For short generations, we generate hidden states and answers with Llama-2 7B and 70B (Touvron et al., 2023b), Mistral 7B (Jiang et al., 2023), and Phi-3 Mini (Abdin et al., 2024), and use DeBERTa-Large (He et al., 2021) to predict entailment. For long generations, we use Llama-2-70B or Llama-3-70B (Meta, 2024) and use GPT-3.5 (Brown et al., 2020) to predict entailment.

**Baselines.** We compare SEPs against the ground truth semantic entropy, accuracy probes supervised with model correctness labels, naive entropy, log likelihood, and the $p(\text{True})$ method of Kadavath et al. (2022). For naive entropy, following Farquhar et al. (2024), we compute the length-normalized average log token probabilities across the same number of generations as for SE. For log likelihood, we use the length-normalized log likelihood of a single model generation. The $p(\text{True})$ method works by constructing a custom few-shot prompt that contains a number of examples – each consisting of a training set input, a corresponding low-temperature model answer, high-temperature model samples, and a model correctness score. Essentially, $p(\text{True})$ treats sampling-based truthfulness detection as an in-context learning task, where the few-shot prompt teaches the model that model answers with high semantic variety are likely incorrect. We refer to Kadavath et al. (2022) for more details. We show additional baselines such as non-linear probing and probing alternative targets like $p(\text{True})$ in Appendix A.

**Linear Probe.** For SEPs and the accuracy probe baseline, we use the logistic regression model from Pedregosa et al. (2011) with default hyperparameters for $L_2$ regularization and the LBFGS optimizer.

**Evaluation.** We evaluate SEPs both in terms of their ability to capture semantic entropy as well as their ability to predict model hallucinations. In both cases, we compute the area under the receiver operating characteristic curve (AUROC), with gold labels given by binarized SE or model accuracy. We confirm the statistical significance of our results and refer to Appendix B for details.

## 6 LLM HIDDEN STATES IMPLICITLY CAPTURE SEMANTIC ENTROPY

This section investigates whether LLM hidden states encode semantic entropy. We study SEPs across different tasks, models, and layers, and compare them to accuracy probes in- and out-of-distribution.

**Hidden States Capture Semantic Entropy.** Figure 2 shows that SEPs are consistently able to capture semantic entropy across different models and tasks. Here, probes are trained on hidden states of the second-last-token for the short-form generation setting. In general, we observe that AUROC values increase for later layers in the model, reaching values between 0.7 and 0.95 depending on the scenario.

**Semantic Entropy Can Be Predicted Before Generating.** Next, we investigate if semantic entropy can be predicted before even generating the output. Similar to before, Fig. 3 shows AUROC values for predicting binarized semantic entropy from the SEP probes. Perhaps surprisingly (although in line with related work, cf. Section 2), we find that SEPs can capture semantic entropy even before generation. SEPs consistently achieve good AUROC values, with performance slightly below the SLT experiments in Fig. 2. The TBG variant provides even larger cost savings than SEPs already do, as it allows us to quantify uncertainty before generating any novel tokens, i.e. with a single forward pass through the model. This could be useful in practice, for example, to refrain from answering queries for which semantic uncertainty is high.

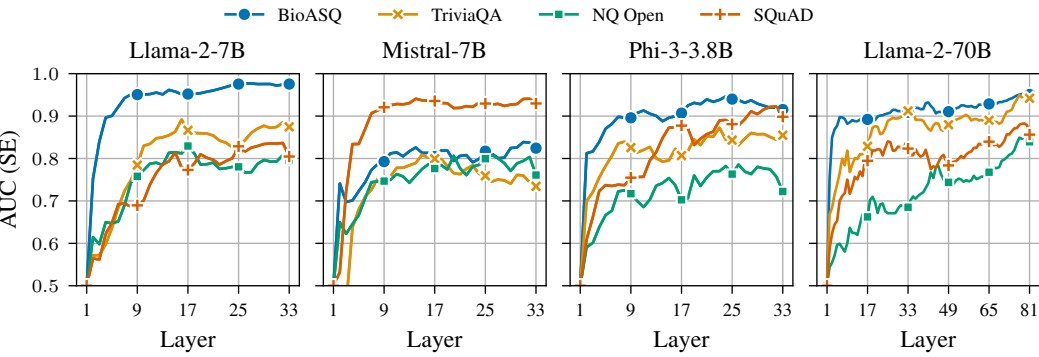

Figure 2: Semantic Entropy Probes (SEPs) achieve high fidelity for predicting semantic entropy. Across datasets and models, SEPs are consistently able to capture semantic entropy from hidden states of mid-to-late layers. Short generation scenario with probes trained on second-last token (SLT).

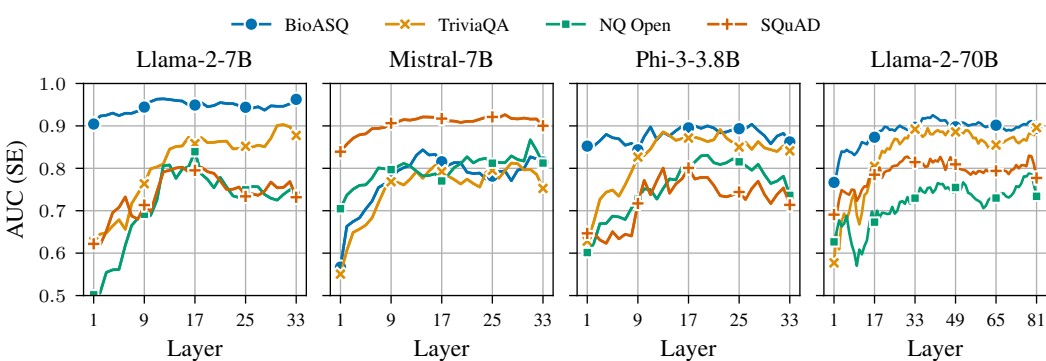

Figure 3: Semantic entropy can be predicted from the hidden states of the last input token, without generating any novel tokens. Short generations with SEPs trained on the token before generating (TBG).

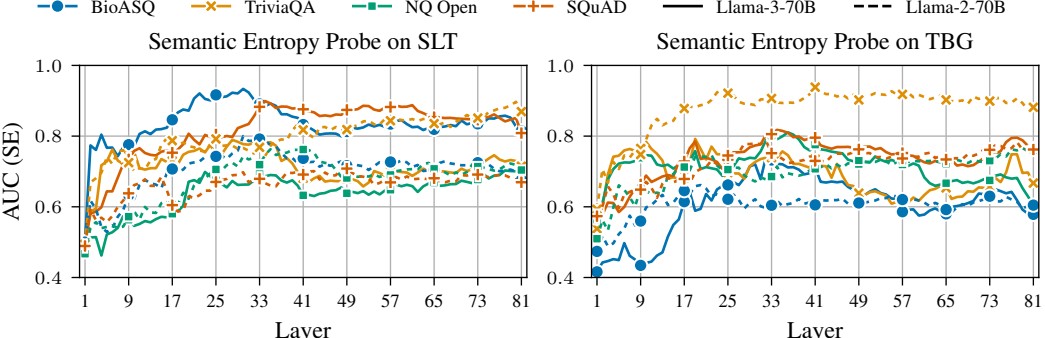

Figure 4: SEPs successfully capture semantic entropy in Llama-2-70B and Llama-3-70B for long generations across layers and for both SLT and TBG token positions.

AUROC values for Llama-2-7B on BioASQ, in both Fig. 2 and Fig. 3, reach very high values, even for early layers. We investigated this and believe it is likely related to the particularities of BioASQ. Concretely, it is the only of our tasks to contain a significant number of yes-no questions, which are generally associated with lower semantic entropy as the possible number of semantic meanings in outcome space is limited. For a model with relatively low accuracy such as Llama-2-7B, simply identifying whether or not the given input is a yes-no question, will lead to high AUROC values.

**SEPs Capture Semantic Uncertainty for Long Generations.** While experiments with short generations are popular even in the recent literature (Kuhn et al., 2023; Kadavath et al., 2022; Duan et al., 2023; Cole et al., 2023; Chen & Mueller, 2023), this scenario is increasingly disconnected from popular use cases of LLMs as conversational chatbots. In recognition of this, we also study our probes in a long-form generation setting, which increases the average length of model responses from ~15 characters in the short-length scenario to about ~100 characters.

Figure 4 shows that, even in the long-form setting, SEPs are able to capture semantic entropy well in both the second-last-token and token-before-generation scenarios for Llama-2-70B and Llama-3-70B. Compared to the short-form generation scenario, we now observe more often that AUROC values peak for intermediate layers. This makes sense as hidden states closer to the final layer will likely be preoccupied with predicting the next token. In the long-form setting, the next token is more often unrelated to the semantic uncertainty of the overall answer, and instead concerned with syntax or lexis.

**Counterfactual Context Addition Experiment.** To confirm that SEPs capture SE rather than relying on spurious correlations, we perform a counterfactual intervention experiment for Llama-2-7B on TriviaQA. For each input question of TriviaQA, the dataset contains a "context", from which the ground truth answer can easily be predicted. We usually exclude this context, because including it makes the task too easy. However, for the purpose of this experiment, we add the context and study how this affects SEP predictions.

Figure 5 shows a kernel density estimate of the distribution over the predicted probability for high semantic entropy, $p$(high SE), for

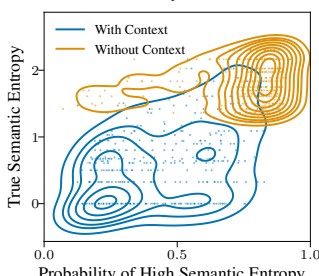

Fig. 5: SEPs capture drop in SE due to added context.

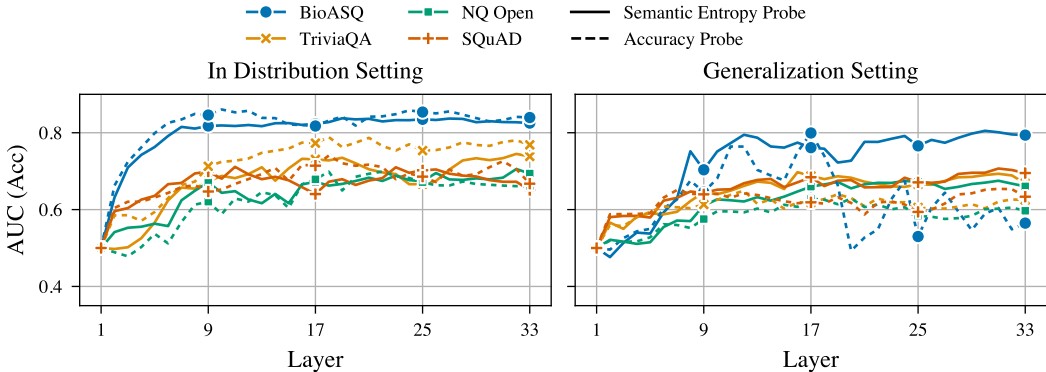

Figure 6: SEPs predict model hallucinations better than accuracy probes when generalizing to unseen tasks. In-distribution, accuracy probes perform better. Short generation setting with Llama-2-7B, SEPs trained on the second-last-token (SLT). For the generalization setting, probes are trained on all tasks except the one that we evaluate on.

Llama-2-7B on the TriviaQA dataset with context (blue) and without context (orange) in the short generation setting using the SLT. Without context, the distribution for $p(\text{high SE})$ from the SEP is concentrated around 0.9. However, as soon as we provide the context, $p(\text{high SE})$ decreases, as shown by the shift in distribution. As the task becomes much easier – accuracy increases from 26% to 78% – the model becomes more certain – ground truth SE decreases from 1.84 to 0.50. This indicates SEPs accurately capture model behavior for the context addition experiment, with predictions for $p(\text{high SE})$ following ground truth SE behavior, despite never being trained on inputs with context.

## 7 SEPs Are Cheap and Reliable Hallucination Detectors

In this section, we explore the use of SEPs to predict hallucinations, comparing them to accuracy probes and other baselines. Crucially, we also evaluate probes in a challenging generalization setting, testing them on tasks that they were not trained for. This setup is much more realistic than evaluating probes in-distribution, as, for most deployment scenarios, inputs will rarely match the training distribution exactly. The generalization setting does not affect semantic entropy, naive entropy, or log likelihood, which do not rely on training data. While $p(\text{True})$ does rely on a few samples for prompt construction, we find its performance is usually unaffected by the task origin of the prompt data.

Figure 6 shows both in-distribution and generalization performance of SEPs and accuracy probes across different layers for Llama-2-7B in a short-form generation setting trained on the SLT. In-distribution, accuracy probes outperform SEPs across most layers and tasks, with the exception of NQ Open. In Table 1, we compare the average difference in AUROC between SEPs and accuracy probes for predicting model hallucinations, training probes on a concatenation of high-performing layers for both probe types (see Appendix B). We find that SEPs and accuracy probes perform similarly on in-distribution data across models. We report unaggregated results in Fig. A.8. The performance of SEPs here is commendable: SEPs are trained without any ground truth answers or accuracy labels, and yet, can capture truthfulness. To the best of our knowledge, SEPs may be the best unsupervised method for hallucination detection even in-distribution, given problems of other unsupervised methods for truthfulness prediction (Farquhar et al., 2023).

Tab. 1: $\Delta$AUROC (x100) of SEPs and acc. probes over tasks in-distribution. Avg $\pm$ std error, (S)hort- and (L)ong-form gens.

| Model | SEP $-$ Acc Pr. |
|---|---|
| Mistral-7B (S) | $2.8 \pm 1.4$ |
| Phi-3-3.8B (S) | $2.1 \pm 0.8$ |
| Llama-2-7B (S) | $-0.5 \pm 2.6$ |
| Llama-2-70B (S) | $1.3 \pm 0.7$ |
| Llama-2-70B (L) | $-1.9 \pm 7.5$ |
| Llama-3-70B (L) | $-2.0 \pm 2.1$ |

However, when evaluating probe generalization to new tasks, SEPs show their true strength. We evaluate probes in a leave-one-out fashion – evaluating on all datasets except one, which we train on. As shown in Fig. 6 (right), SEPs consistently outperform accuracy probes across various layers and tasks for short-form generations in the generalization setting. For BioASQ, the difference is particularly large. SEPs clearly generalize better to unseen tasks than accuracy probes. In Table 2 and Fig. 7, we report results for more models, taking a representative set of high-performing layers for

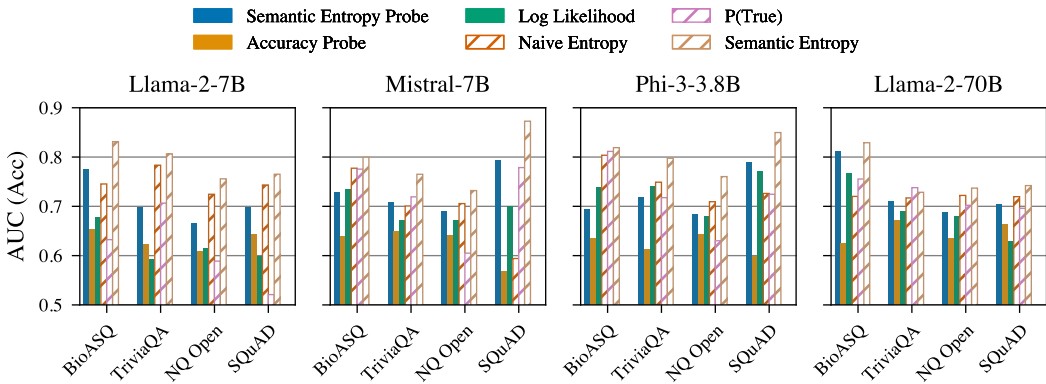

Figure 7: SEPs generalize better to new tasks than accuracy probes across models and tasks. They approach, but do not match, the performance of other, 10x costlier baselines (hatched bars). Short generation setting, SLT, performance for a selection of representative layers.

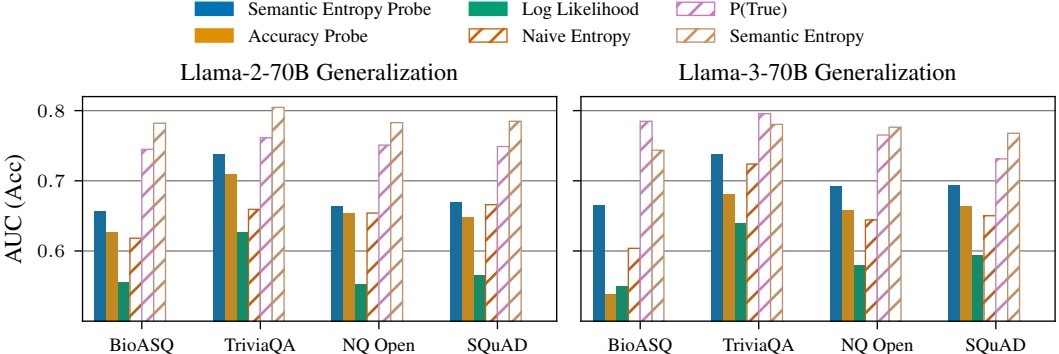

Figure 8: Semantic entropy probes outperform accuracy probes for hallucination detection in the long-form generation generalization setting with both Llama-2-70B and Llama-3-70B.

both probe types, and Fig. A.7 shows results for Mistral-7B across layers. We again find that SEPs generalize better than accuracy probes to novel tasks. We additionally compare to the sampling-based semantic entropy, naive entropy, and $p(\text{True})$ methods. While SEPs cannot match the performance of these methods, it is important to note the significantly higher cost these baselines incur, requiring 10 additional model generations, whereas SEPs and accuracy probes operate on single generations.

We further evaluate SEPs for long-form generations. As shown in Fig. 8, SEPs outperform accuracy probes for Llama-2-70B and Llama-3-70B in the generalization setting. We also provide in-distribution results for long generations with both models in Figs. A.9 and A.10. Both results confirm the trend discussed above. Overall, our results clearly suggest that SEPs are the best choice for cost-effective uncertainty quantification in LLMs, especially if the distribution of the query data is unknown.

Tab. 2: $\Delta$AUROC (x100) of SEPs over acc. probes for task generalization. Avg $\pm$ std error, (S)hort- and (L)ong-form gens.

| Model | SEP $-$ Acc Pr. |
|---|---|
| Mistral-7B (S) | $10.5 \pm 3.5$ |
| Phi-3-3.8B (S) | $9.9 \pm 2.9$ |
| Llama-2-7B (S) | $7.7 \pm 1.3$ |
| Llama-2-70B (S) | $7.9 \pm 3.0$ |
| Llama-2-70B (L) | $2.2 \pm 0.4$ |
| Llama-3-70B (L) | $6.2 \pm 1.9$ |

## 8 DISCUSSION, FUTURE WORK, AND CONCLUSIONS.

**Discussion.** On inputs from unseen tasks, our experiments show that SEPs are better predictors of hallucinations than accuracy probes. In this discussion, we would like to offer some intuition as to why we believe this is the case. Empirically, we find that SE is a quantity that can be reliably and consistently predicted from the model hidden states. We believe that SEPs learn to extract this signal and therefore generalize well to new tasks. Accuracy probes, on the other hand, may instead learn less robust features that generalize worse to new tasks.

The reason for this, ultimately, may be that accuracy depends on an *external* labeling process, and there is therefore no general mechanism to reliably predict accuracy from model-internal hidden states

across a large number of queries. We believe that the challenges from associating hard accuracy labels to linguistic statements lead to accuracy probes not learning robust features of model uncertainty and instead learning features that rely on spurious correlations between model predictions and accuracy on a particular dataset. Concretely, we rarely but consistently observe outlier datapoints where confident model predictions are labeled as 'inaccurate' because the model does not exactly match the gold answer, for example, answering a slightly different question or answering at the wrong level of granularity. For SEPs, such datapoints are trouble-free: the probe will learn to extract the SE signal in latent space, which is reliably aligned with the SE labels used to supervise the probes. For accuracy probes, however, we hypothesize that such datapoints are highly problematic: for them, the 'SE signal' in latent space is not a reliable enough predictor of model accuracy. Thus, the accuracy probe will learn to rely on other features, for example, a feature that identifies inputs for which there is a discrepancy between model answers and expected gold labels on a particular dataset. As labeling procedures differ between datasets, the accuracy probe struggles to generalize to new tasks. For example, it may be that the model confidently replies to 'historical questions' by giving only a (correct) year when the gold answer on the training dataset expects a full date. When the generalization dataset then requires only years, the features of the accuracy probes fail to generalize while SEPs perform well.

In summary, a possible explanation for the gap in OOD generalization is that accuracy probes capture model correctness in a way that is *specific* to the training dataset. They may latch on to discriminative features for model correctness that relate to the task at hand but do not generalize, such as identifying a knowledge domain where accuracy is high or low, but which rarely occurs outside the training data. Conversely, SEPs capture inherent model uncertainty which generalizes well to new tasks.

**Limitations.** While they are extremely cheap to compute, SEPs do not match the performance of sampling-based methods such as full SE. Further, given that we need to compute SE for the training set to train SEPs, they are computationally advantageous only when the number of test queries is larger than the training set – which should be a fair assumption for most, but certainly not all, deployment scenarios. Lastly, like other probing approaches, SEP require full access to model internals and cannot be computed in black-box scenarios.

**Future Work.** We believe it should be possible to further close the performance gap between sampling-based approaches, such as semantic entropy, and SEPs. One avenue to achieve this could be to increase the scale of the training datasets used to train SEPs. In this work, we relied on established QA tasks to train SEPs to allow for easy comparison to accuracy probes. However, future work could explore training SEPs on unlabelled data, such as inputs generated from another LLM or natural language texts used for general model training or finetuning. This could massively increase in the amount of training data for SEPs, which should improve probe accuracy, and also allow us to explore other more complex probing techniques that require more training data.

**Conclusions.** We have introduced semantic entropy probes (SEPs): linear probes trained on the hidden states of LLMs to predict semantic entropy (Kuhn et al., 2023), an effective measure of uncertainty for free-form LLM generations. We find that the hidden states of LLMs implicitly capture semantic entropy across a wide range of scenarios. SEPs are able to predict semantic entropy consistently, and, importantly, they detect model hallucinations more effectively than probes trained directly for accuracy prediction when testing on novel inputs from a different distribution than the training set – despite not requiring any ground truth model correctness labels. Semantic uncertainty probing, both in terms of model interpretability and practical applications, is an exciting avenue for further research.

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

## A  ADDITIONAL RESULTS

**Model Task Accuracies.**  We report the accuracies achieved by the models on the various datasets used in this work in Table 3.

Table 3: Task accuracy of models across datasets, in (L)ong- and (S)hort-form generation settings.

| Model | BioASQ (%) | TriviaQA (%) | NQ Open (%) | SQuAD (%) |
|---|---|---|---|---|
| **Llama-3-70B (L)** | 67.2 | 88.5 | 61.2 | 46.0 |
| **Llama-2-70B (L)** | 60.3 | 85.0 | 58.3 | 43.9 |
| **Llama-2-70B (S)** | 48.4 | 75.7 | 49.5 | 31.4 |
| **Llama-2-7B (S)** | 43.3 | 64.8 | 38.3 | 23.5 |
| **Mistral-7B (S)** | 39.3 | 52.3 | 28.3 | 20.7 |
| **Phi-3-3.8B (S)** | 45.5 | 48.3 | 26.1 | 24.3 |

**Predicting Model Correctness from Hidden States.**  Figures A.1 and A.2 give additional results that show we can predict model correctness from hidden states using SEPs trained on the second-last-token (SLT) or token-before-generating (TBG) in the short-form in-distribution scenario across models and tasks. In Figs. A.3 and A.4, we further demonstrate that accuracy probes also perform similarly when trained on the SLT or TBG in the short-form in-distribution scenario across models and tasks.

**Predicting Correctness vs. Semantic Entropy.**  Figures A.5 and A.6 show that predicting semantic entropy from hidden states is generally easier than directly predicting model correctness, suggesting that semantic entropy is implicitly encoded in the hidden states.

**Additional Comparisons to Baselines.**  In, Fig. A.7 we additionally report results comparing SEPs to accuracy probes across layers for Mistral-7B for the in-distribution and generalization settings. In Fig. A.8, we compare the performance of SEPs to baselines for the in-distribution setting across models and datasets, finding that SEPs and accuracy probes perform similarly, with SEPs performing slightly better for 3 out of 5 models. In Figs. A.9 and A.10 we report in- and out-of-distribution results for Llama-2-70B and Llama-3-70B in the long-form generation setting.

We also includes four token-based baselines (i.e., average or minimum of token probabilities, and log likelihoods of SLT or TBG tokens), linear regression (LR) probes for continuous SE, and non-linear (MLP) probes for accuracy and SE. For both the linear regression and MLP models, training targets are normalized semantic entropy values, and the predictions are subsequently converted into probabilities using a `sigmoid` function. In total, we have evaluated 13 baselines in addition to SEPs.

Each non-linear probe is trained using an Adam optimizer, with a learning rate of 5e-4, over 50 epochs with a batch size of 32. The loss function used is mean squared error (MSE). To mitigate overfitting, we apply batch normalization after each linear layer and use a dropout rate of 0.5 following each non-linearity.

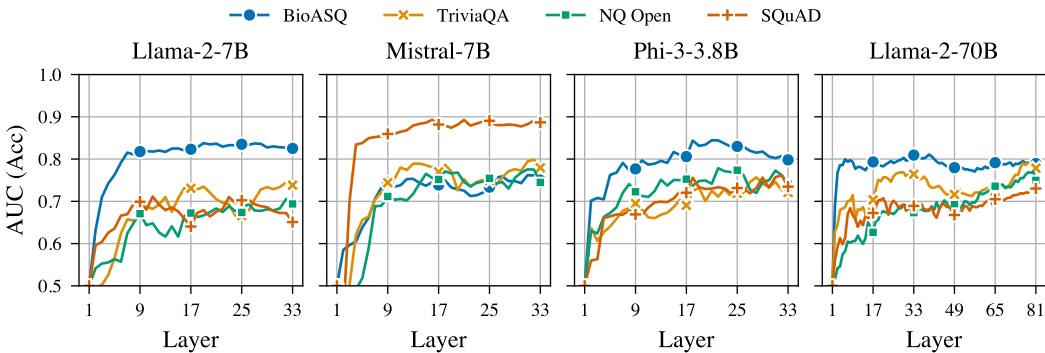

Figure A.1: Semantic Entropy Probes (SEPs) capture model hallucinations. Short generations with SEPs trained on the hidden states of the model at the second-last-token (SLT).

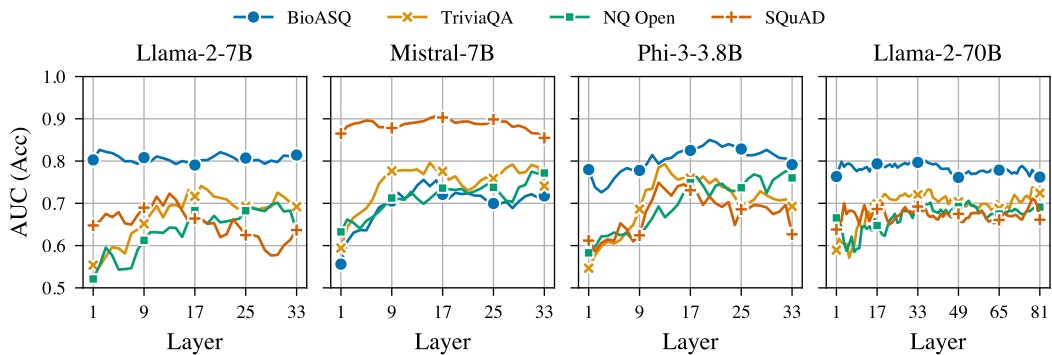

Figure A.2: Semantic Entropy Probes (SEPs) capture model hallucinations. Short generations with SEPs trained on the hidden states of the model at the token-before-generation (TBG).

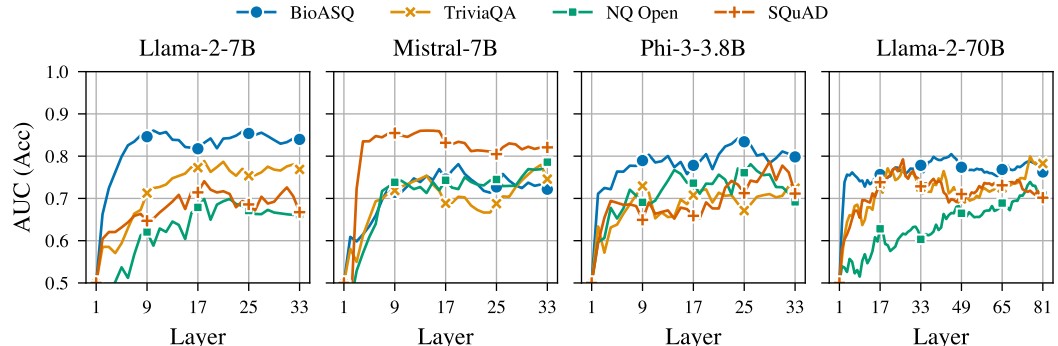

Figure A.3: Accuracy probes for in-distribution short-form generation trained on the second-last-token (SLT).

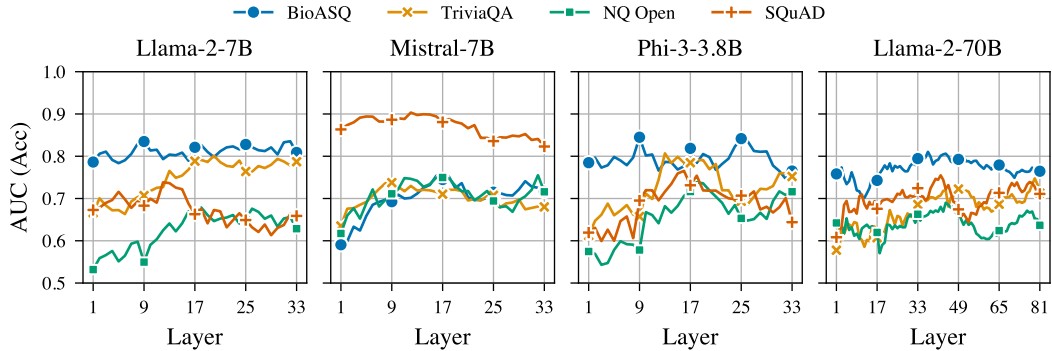

Figure A.4: Accuracy probes for in-distribution short-form generation trained on the token-before-generation (TBG).

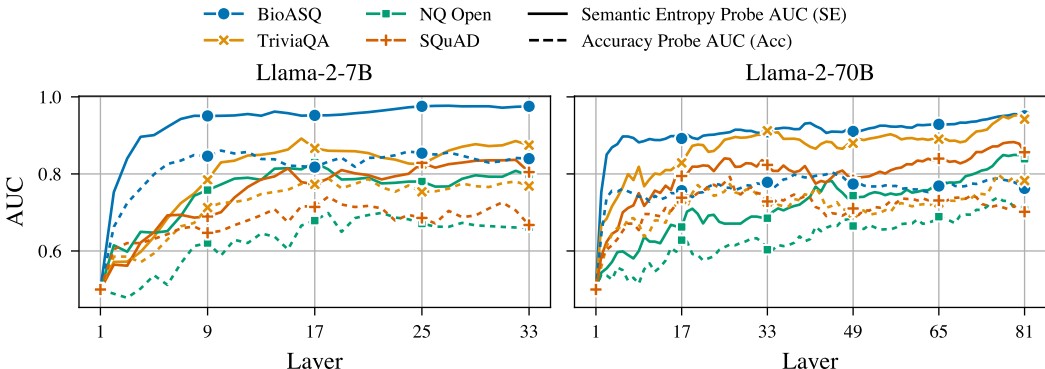

Figure A.5: Predicting semantic entropy from hidden states with SEPs works better than predicting accuracy from the hidden states with accuracy probes. Llama-2-7B and 70B in the short generation setting with probes trained on hidden states of the SLT, evaluated in-distribution.

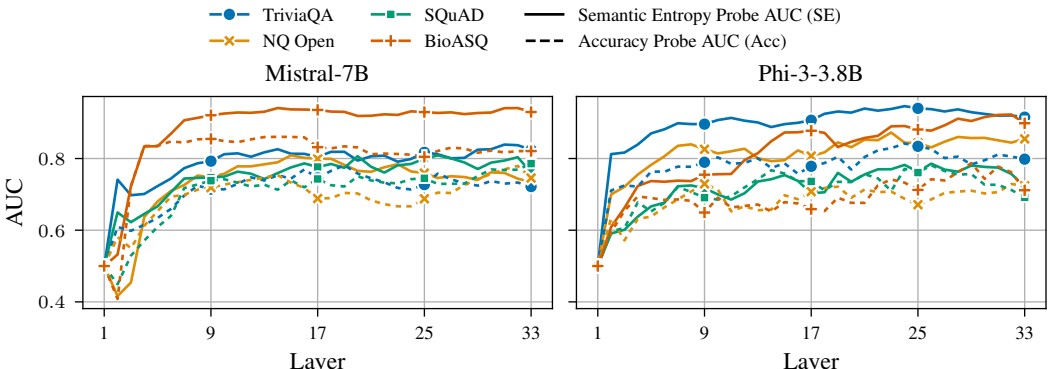

Figure A.6: Predicting semantic entropy from hidden states with SEPs works better than predicting accuracy from the hidden states with accuracy probes. Mistral-7B and Phi-3 Mini in short generation setting with probes trained on hidden states of the SLT, evaluated in-distribution.

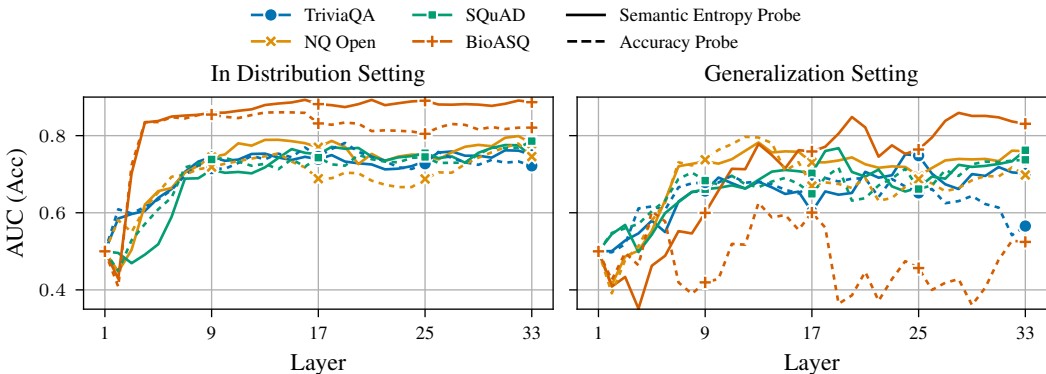

Figure A.7: SEPs predict model hallucinations better than accuracy probes when generalizing to unseen tasks (right). In-distribution, accuracy probes have comparable performance (left). Mistral-7B in the short generations setting with probes trained hidden states from the SLT. For the generalization setting, probes are trained on all tasks except the one that we evaluate on.

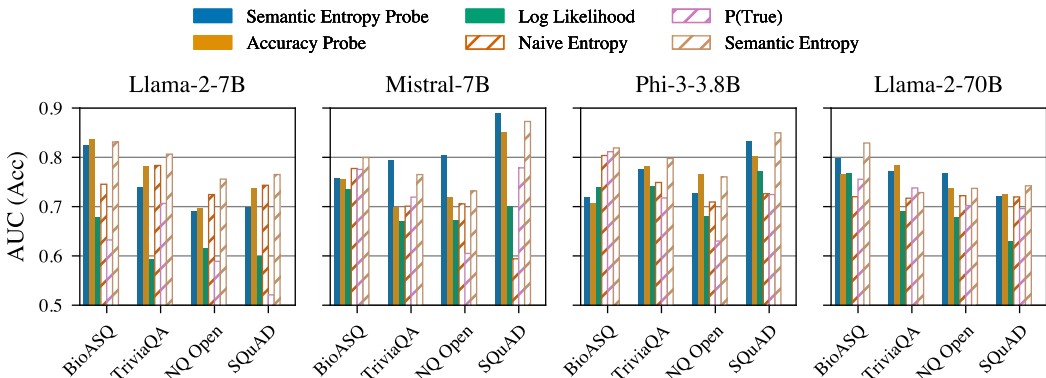

Figure A.8: Short generation performance for the in-distribution setting across models compared to baseline methods. Hatched bars indicate more computationally expensive methods.

We show the comparisons of performances between SEPs and the additional baselines in Fig. A.11. The findings reveal that SEPs consistently outperform all baselines by a significant margin in the generalization setting, further confirming their reliability as a tool for detecting hallucinations.

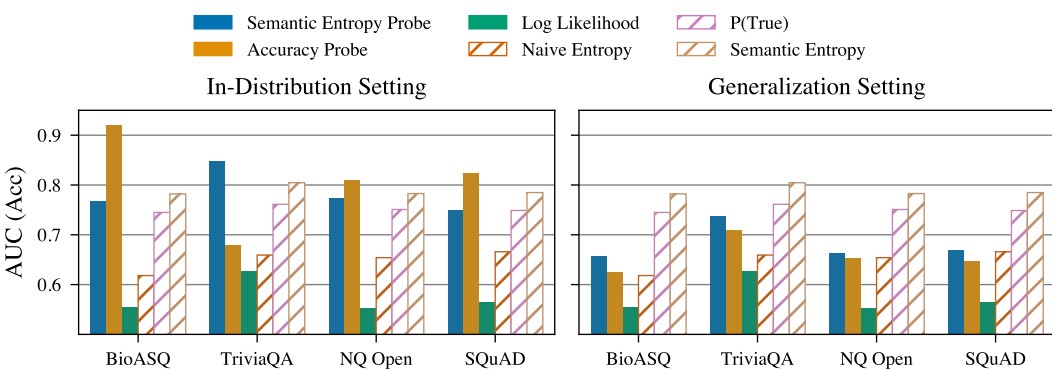

Figure A.9: Semantic entropy probes outperform accuracy probes for hallucination detection in the long-form generation generalization setting with Llama-2-70B. In-distribution, accuracy probes sometimes outperform and sometimes underperform. Probes cannot match the performance of the significantly more expensive baselines.

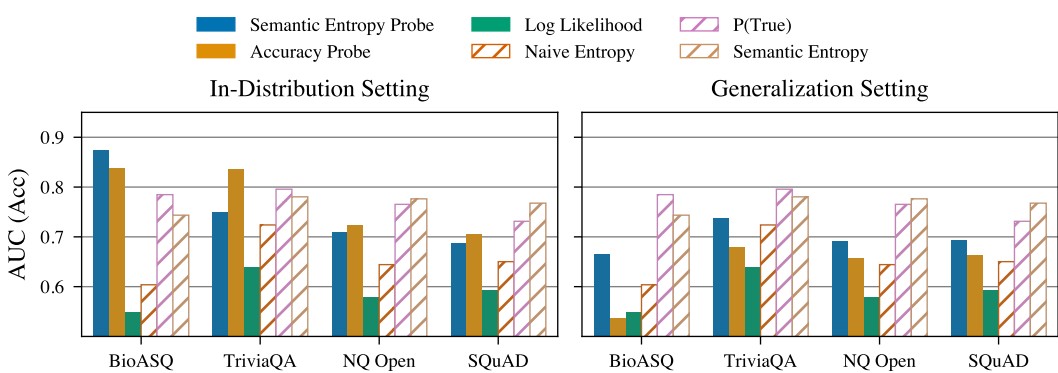

Figure A.10: Semantic entropy probes outperform accuracy probes for hallucination detection in the long-form generation generalization setting with Llama-3-70B. In-distribution, accuracy probes often outperform SEPs. Probes cannot match the performance of much more expensive baselines.

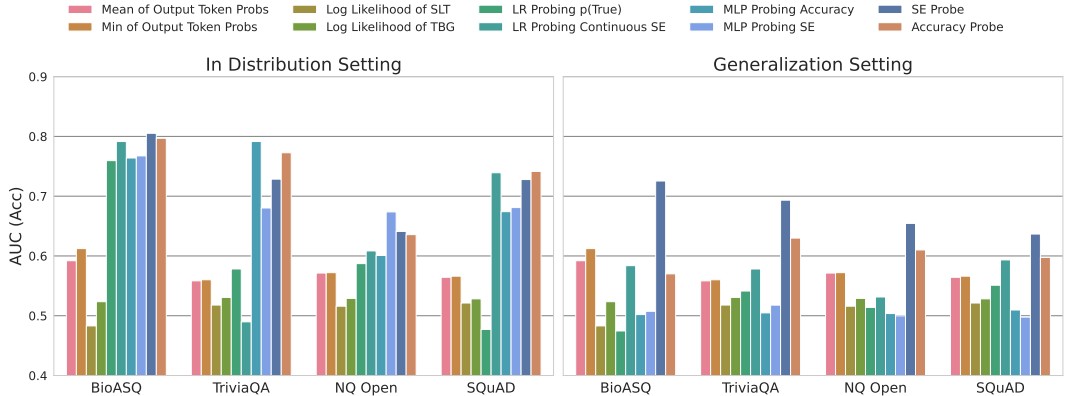

Figure A.11: SEPs outperform additional baselines, in particular in generalization scenarios. These baselines include token-based baselines, probes for $p(\text{True})$, and non-linear methods like MLPs. The results highlight the efficacy of SEPs in the short-generation setting on the Llama-2-7B model.

**Hidden State Alternatives.** In addition to investigating the performance of probes on the hidden states, we study whether residual stream or MLP outputs can also be used for semantic entropy prediction. Figure A.12 shows that probing the hidden states results in consistently higher performance.

**Different Binarization Procedures.** In addition to the "best split" procedure discussed in Section 4 and used in all of our experiments, we here explore the performance of a simple "even split" alternative,

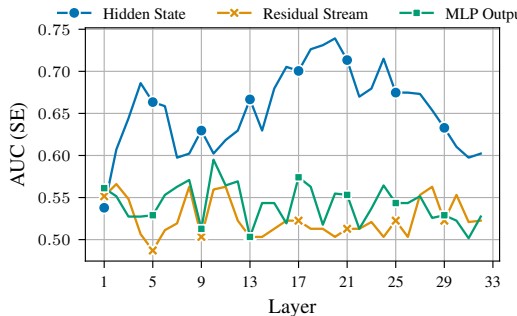

Figure A.12: Probing different model components for SEPs. The hidden states are more predictive than residual streams and MLP outputs. TriviaQA, Llama-2-7B, in-distribution, short-form generations, SLT.

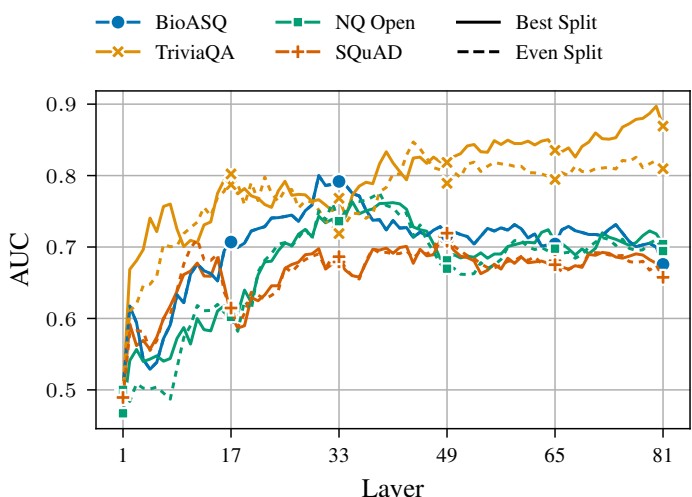

Figure A.13: Comparing binarization methods for semantic entropy. Our "best split" procedure slightly outperforms the "even split" strategy, although SEPs do not appear overly sensitive to the binarization procedure. Long-form generations for Llama-2-70B, SLT, in-distribution.

which splits semantic entropy into high and low classes such that there are an equal number samples in both classes. Figure A.13 shows that performance is similar, with our optimal splitting procedure slightly outperforming the even split ablation. For illustration purposes, Fig. A.14 shows the behavior of the best split objective Eq. (2) across different thresholds. Figure A.15 further shows the best-split objective for Llama-2-7B aross tasks. We have also explored a "soft labelling" strategy as an alternative to hard binarization, for which we obtain soft labels by transforming raw semantic entropies into probabilities with a sigmoid function centered around the best-split threshold, and then train SEPs on the resulting soft labels. Early results did not improve performance.

**Computational Costs for SEPs and SE.** We here give a demonstration of the real-world computational cost-savings associated with computing SEP instead of full SE. We find that, on average, computing SE for 1,000 TriviaQA samples for Llama-2-7B takes $168 \pm 19.11$ minutes. In contrast, the computation of SEP predictions requires only $7.13 \pm 0.14$ seconds in the same setting. Consequently, SEPs are able to speed up hallucination detection by *1,413-fold* compared to full SE.

**Rejection-Accuracy Curve for SEPs.** Figure A.16 shows rejection-accuracy curves (Farquhar et al., 2024) for Llama-2-7B and SEPs. These curves show that SEPs can improve the overall accuracy of the model by refusing to predict on those inputs where SEPs predict the highest uncertainty.

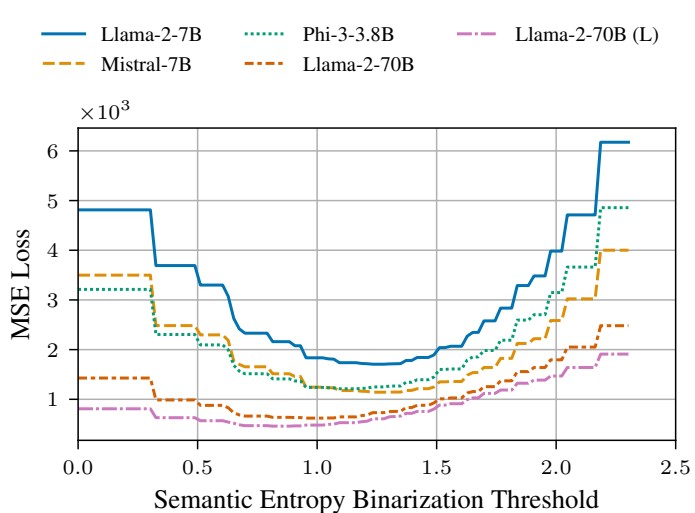

Figure A.14: MSE of the best-split objective Eq. (2) for different binarization thresholds $\gamma$ for models in either short-form generation or (L)ong-form generation settings (SLT).

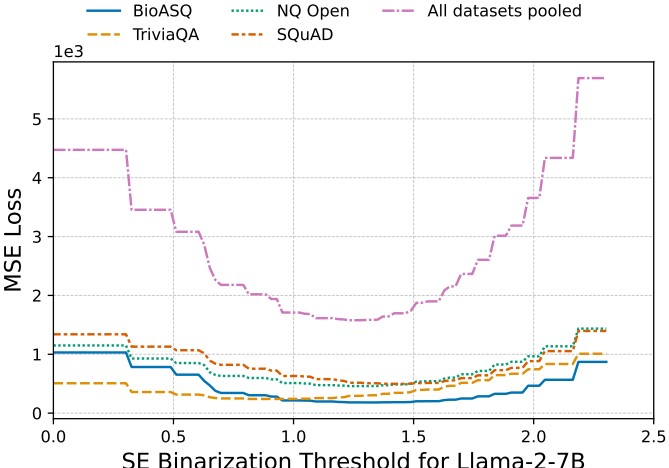

Figure A.15: MSE of the best-split objective Eq. (2) for individual datasets at different binarization thresholds. Evaluated on Llama-2-7B in the short-form generation setting (SLT).

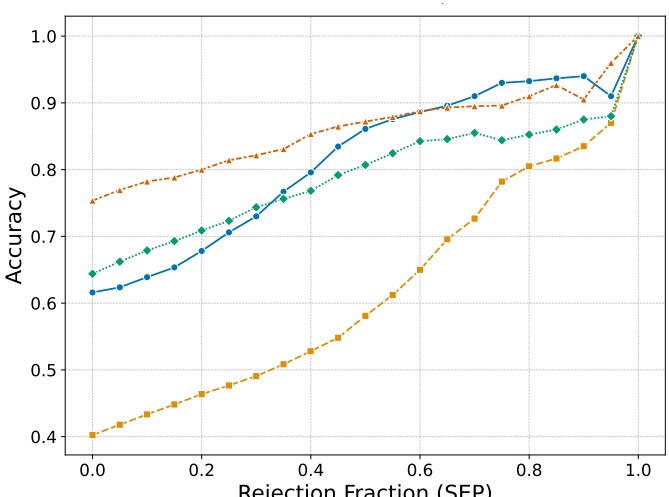

Figure A.16: Accuracy scores of the Llama-2-7B model across SEP-based rejection fractions ranging from 0 to 1. The model prediction is rejected at threshold $m$ when the SEP outputs of the model's generation is below the fraction $m$ of all probe predictions.

# B  EXPERIMENT DETAILS

Here we provide additional details to reproduce the experiments of the main paper.

## B.1  MULTI-SEED EXPERIMENTS

We conduct both layer-concatenated and layer-wise experiments for in-distribution (ID) and out-of-distribution (OOD) hallucination detection, utilizing random seeds 0, 1, 2, 3, and 42. We report the standard errors of test AUROC scores over seeded runs for probing methods and the bootstrapping errors for non-probing baselines in Figs. B.1 and B.2 and Table 4. Our results clearly indicate that SEPs outperform accuracy probes with statistical significance in the generalization setting and that SEPs perform similar to accuracy probes in-distribution.

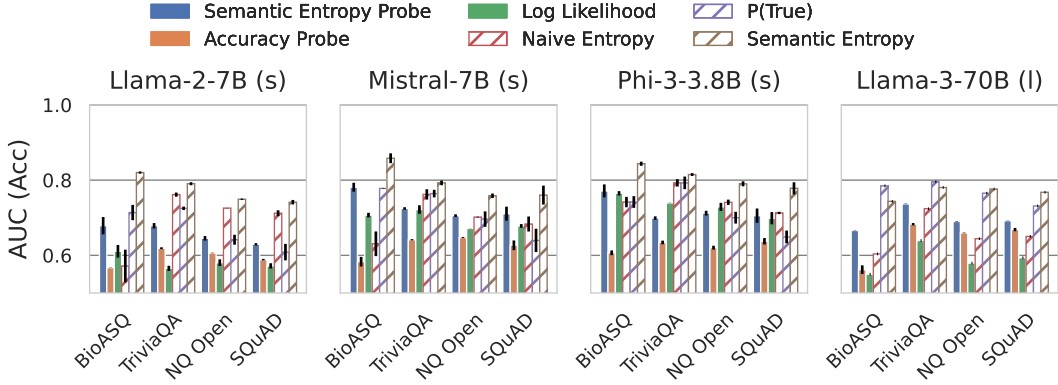

Figure B.1: A reproducibility study confirms SEP's superior performance compared to other computationally cheap baselines, namely accuracy probes and token log likelihoods. For probing methods, we report means and standard errors (which are sometimes too small to be visible) across 5 random seeds. We show bootstrapped errors for non-probing baselines. Results for generalization setting with both (s)hort-and (l)ong-form generations, with probes trained on the second-last-token (SLT).

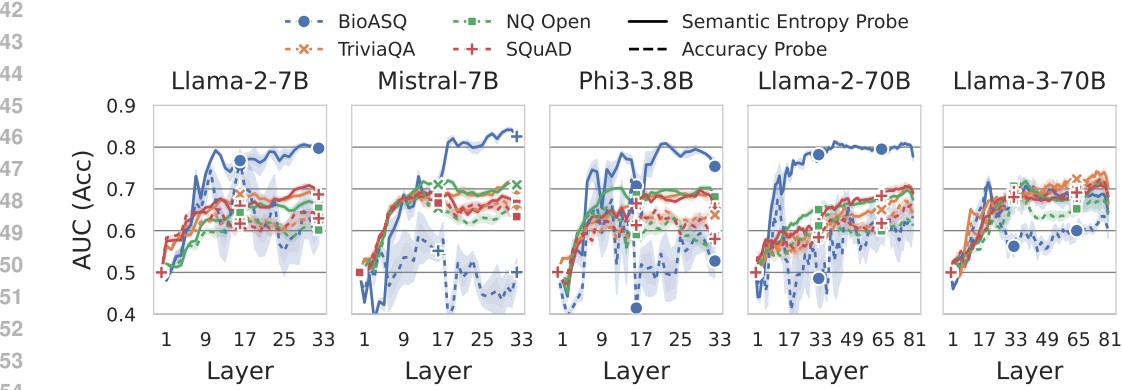

Figure B.2: Layer-wise results from the reproducibility study in Figure B.1. SEPs consistently generalize better than accuracy probes across various probing locations. The figure shows means and standard errors over five random seeds. Probes were trained on the second-last token (SLT) in short-form generation settings, except for Llama-3-70B, which used the long-form setting.

## B.2 PROMPT TEMPLATES

We use the following prompt templates across experiments.

For long-form generations, we use the following prompt template:

```
Answer the following question in a single brief but complete sentence.
Question: [query question]
Answer:
```

For short-form generations, we adjust the instruction and additionally provide 5 demonstration examples with short ground truth answers, to elicit a short answer from the model:

```
Answer the following question as briefly as possible.
Question: [example question 1]
Answer:   [example answer 1]
...
Question: [example question 5]
Answer:   [example answer 5]
Question: [query question]
Answer:
```

Finally, for the counterfactual context addition experiment, we prepend the context, prior to the question:

```
Context: [query context]
Question: [query question]
Answer:
```

## B.3 SEMANTIC ENTROPY CALCULATION

We compute semantic entropy with $N = 10$ generations sampled at temperature $T = 1.0$ and using default values of top-p ($p = 0.9$) and top-K ($K = 50$).

For short-form generations, we predict entailment using DeBERTa-Large (He et al., 2021) and assess model accuracy via the SQuAD F1 score.

For long-form generations, we predict entailment with GPT-3.5 (Brown et al., 2020) and the following prompt:

```
Here are two possible answers:
```

Table 4: Table metrics for the test performances of SEP and Accuracy Probes (Acc. P.) in generalization (OOD) and in-distribution (ID) settings for models making (S)hort-form and (L)ong-form generations, with standard errors (subscripted in brackets) over 5 seeded runs. Bold values indicate 95% significant differences where $|\text{SEP} - \text{Acc. P.}| > 2 \times \text{Std. Err.}$.

**Out-of-Distribution (OOD) Results**

|  | BioASQ | TriviaQA | NQ Open | SQuAD |
|---|---|---|---|---|
| **Llama-2-7B$_{(S)}$** | | | | |
| SEP | **0.6789**$_{(0.0229)}$ | **0.6787**$_{(0.0071)}$ | **0.6453**$_{(0.0060)}$ | **0.6294**$_{(0.0030)}$ |
| Acc. P. | 0.5674$_{(0.0004)}$ | 0.6190$_{(0.0024)}$ | 0.6066$_{(0.0005)}$ | 0.5887$_{(0.0019)}$ |
| **Mistral-7B$_{(S)}$** | | | | |
| SEP | **0.6761**$_{(0.0070)}$ | **0.7352**$_{(0.0352)}$ | **0.7050**$_{(0.0122)}$ | **0.7010**$_{(0.0043)}$ |
| Acc. P. | 0.6459$_{(0.0025)}$ | 0.5543$_{(0.0111)}$ | 0.6498$_{(0.0052)}$ | 0.6412$_{(0.0071)}$ |
| **Phi-3-3.8B$_{(S)}$** | | | | |
| SEP | **0.6967**$_{(0.0068)}$ | **0.7624**$_{(0.0076)}$ | **0.6907**$_{(0.0007)}$ | **0.7159**$_{(0.0016)}$ |
| Acc. P. | 0.6207$_{(0.0183)}$ | 0.5891$_{(0.0212)}$ | 0.6205$_{(0.0094)}$ | 0.6191$_{(0.0034)}$ |
| **Llama-3-70B$_{(L)}$** | | | | |
| SEP | **0.6664**$_{(0.0005)}$ | **0.7367**$_{(0.0006)}$ | **0.6898**$_{(0.0008)}$ | **0.6920**$_{(0.0004)}$ |
| Acc. P. | 0.5682$_{(0.0126)}$ | 0.6833$_{(0.0039)}$ | 0.6601$_{(0.0009)}$ | 0.6699$_{(0.0053)}$ |

**In-Distribution (ID) Results**

|  | BioASQ | TriviaQA | NQ Open | SQuAD |
|---|---|---|---|---|
| **Llama-2-7B$_{(S)}$** | | | | |
| SEP | 0.7757$_{(0.0085)}$ | 0.7297$_{(0.0175)}$ | 0.6736$_{(0.0248)}$ | 0.6605$_{(0.0236)}$ |
| Acc. P. | 0.7722$_{(0.0029)}$ | **0.7764**$_{(0.0157)}$ | 0.6750$_{(0.0274)}$ | **0.6888**$_{(0.0196)}$ |
| **Mistral-7B$_{(S)}$** | | | | |
| SEP | 0.7288$_{(0.0186)}$ | 0.8337$_{(0.0057)}$ | 0.7846$_{(0.0181)}$ | 0.7267$_{(0.0423)}$ |
| Acc. P. | 0.7317$_{(0.0118)}$ | 0.8023$_{(0.0163)}$ | 0.7665$_{(0.0163)}$ | 0.7227$_{(0.0321)}$ |
| **Phi-3-3.8B$_{(S)}$** | | | | |
| SEP | 0.7309$_{(0.0208)}$ | 0.7949$_{(0.0097)}$ | 0.7650$_{(0.0101)}$ | 0.7331$_{(0.0361)}$ |
| Acc. P. | 0.7074$_{(0.0176)}$ | 0.7780$_{(0.0085)}$ | 0.7648$_{(0.0152)}$ | 0.7124$_{(0.0245)}$ |
| **Llama-3-70B$_{(L)}$** | | | | |
| SEP | 0.7558$_{(0.0323)}$ | 0.7230$_{(0.0351)}$ | 0.7166$_{(0.0178)}$ | 0.7270$_{(0.0197)}$ |
| Acc. P. | 0.7636$_{(0.0258)}$ | **0.7998**$_{(0.0402)}$ | 0.6989$_{(0.0096)}$ | 0.7202$_{(0.0206)}$ |

```
Possible Answer 1: [model generation a]
Possible Answer 2: [model generation b]
Does Possible Answer 1 semantically entail Possible Answer 2?
Respond with entailment, contradiction, or neutral.
```

To assess the correctness of long-form generations, we prompt GPT-4 (OpenAI, 2023) or GPT-4o[1] as follows

```
We are assessing the quality of answers to the following question:
[query question]
The expected answer is: [ground truth label].
The proposed answer is: [model generation].
Within the context of the question,
does the proposed answer mean the same as the expected answer?
Respond only with yes or no.
Response:
```

[1]We use GPT-4 to evaluate Llama-2-70B but switched to GPT-4o for our more recent experiments on Llama-3-70B given the difference in cost between the two GPT models.

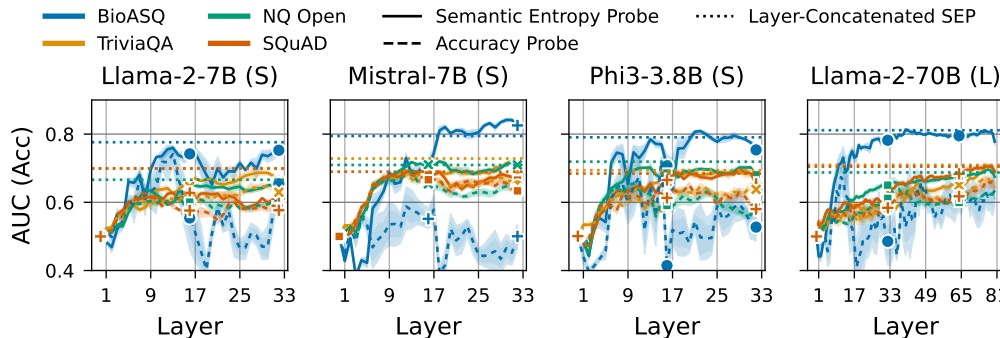

Figure B.3: SEPs trained on concatenated hidden states from multiple adjacent layers generally outperform layer-wise SEPs. Note that we select the set of layers to concatenate globally across all tasks, and, therefore, may not select an optimal set of layers for any given task. Results for both (S)hort- and (L)ong-form generation settings.

### B.4 SEMANTIC ENTROPY PROBES

SEPs are trained on the hidden states, which vary in dimensionality between models. We detail the dimensionality of the hidden states, and number of layers in Table 5.

Table 5: Models properties and selected layers for concatenation for SEPs and (Acc)uracy (P)robe, in (L)ong-form and (S)hort-form generation settings.

| Model Name | No. of Layers | Hidden Dim. | Layers for SEPs | Layers for Acc. P. |
|---|---|---|---|---|
| Llama-3-70B (L) | 80 | 8192 | [76, 77, 78, 79, 80] | [31, 32, 33, 34, 35] |
| Llama-2-70B (L) | 80 | 8192 | [74, 75, 76, 77, 78] | [76, 77, 78, 79, 80] |
| Llama-2-70B (S) | 80 | 8192 | [76, 77, 78, 79, 80] | [75, 76, 77, 78, 79] |
| Llama-2-7B (S) | 32 | 4096 | [28, 29, 30, 31, 32] | [18 ,19 ,20 ,21, 22] |
| Mistral-7B (S) | 32 | 4096 | [28, 29, 30, 31, 32] | [12 ,13 ,14 ,15, 16] |
| Phi-3-3.8B (S) | 32 | 3072 | [21, 22, 23, 24, 25] | [25, 26, 27, 28, 29] |

**Layer Concatenation.** For any aggregate results presented in the main paper or appendix, i.e. any barplots or tables, we report SEP and accuracy probe performance on a representative set of high-performing layers. Concretely, we select a set of adjacent layers and concatenate their hidden states to train both types of probes based on the highest mean AUROC value achieved in the interval (on un-concatenated hidden states) in the in-distribution setting. We report the layers across which we concatenate in Table 5. We present a comparison between SEPs trained on concatenated and individual layers in Fig. B.3.

**Non-Linear Probing.** We configure the hidden dimensions for non-linear probing based on the size of the LLMs, with the default input dimensions matching the shape of $h_p^l(x)$ and the output dimension set to 1. For the 70B models, we use MLP hidden dimensions of $[2048, 512]$ (i.e., 3 layers). For the 7B and 3.8B models, we configure the dimensions as $[2048, 1024, 512]$ (i.e., 4 layers).

**Filtering for Long-form Generations.** In order to provide a clearer signal to the SEP on what constitutes high and low semantic entropy inputs, we filter out training samples with semantic entropy in between the 55% and 80% quantiles for long generations, as we have found this to give a mild increase in performance. Note that this filtering did not improve performance for the accuracy probes, and we report results for the accuracy probes without filtering. We found this filtering to be unnecessary for experiments with Llama-3-70B.

**Training Set Size.** For long-generation experiments, we collect 1000 samples across tasks. For short-generation experiments, we collect 2000 samples of hidden state–semantic entropy pairs across tasks. We match the training set sizes between accuracy probes and SEPs.

### B.5 BASELINES

For the $p$(True) baseline, we construct a few-shot prompt with 10 examples, where each example is formatted as below:

```
Question: [example question 1]
Brainstormed Answers: [model generation a]
[model generation b]
[model generation c]
..
[model generation j]
Possible answer: [greedy model generation]
Is the possible answer:
A) True
B) False
The possible answer is: [A / B depending on correctness of possible answer]
```

We give an illustrative example below for what this could look like in practice:

```
Question: What is the capital of France?
Brainstormed Answers: The capital of France is Paris.
Paris is the capital of France.
It's Paris.
Possible answer: The capital of France is Paris.
Is the possible answer:
A) True
B) False
The possible answer is: A
```

For $p$(True), we obtain the probability of model truthfulness by measuring the token probability of $A$ at the end of the prompt.

### B.6 EVALUATION

To evaluate the performance of the probes in the generalization setting, we employ the following leave-one-out procedure for the aggregate results reported in the barplots and tables.

First, each probe is trained on a single dataset. Then, the trained probes are evaluated on all other datasets in terms of AUROC of detecting hallucinations, excluding the dataset used for training. We then report the mean across all probes evaluated on that specific dataset. This allows us to assess the generalization capability of the probes by measuring their performance on datasets that were not used during the training phase. This scenario is important in practice, as the distribution of the query data will rarely be known.

## C FURTHER RELATED WORK

**Understanding Hidden States.** More generally, probes can be a valuable tool to better understand the internal representations of neural networks like LLMs Alain & Bengio (2017); Belinkov (2021); Marks & Tegmark (2023). Recent work suggests that simple operations on LLM hidden states can qualitatively change model behavior (Subramani et al., 2022; Rimsky et al., 2023; Li et al., 2024) manipulate knowledge (Hernandez et al., 2023), or reveal deceitful intent (MacDiarmid et al., 2024). Previous work has shown that hidden state probes can predict LLM outputs one or multiple tokens ahead with high accuracy (Belrose et al., 2023; Pal et al., 2023).

**Sampling Procedures.** Aichberger et al. (2024) point to potential improvements to uncertainty estimation in LLMs from dedicated sampling schemes that identify token-level contributions to semantic uncertainty.

## D    COMPUTE RESOURCES

We make use of an internal cluster with 24 Nvidia A100 80GB GPUs. We further use GPT 3.5, 4, and 4o via the OpenAI API.

For experiments requiring the use of Llama 70B models, we require 2 A100s to do inference and calculate the hidden states. The smaller models require only a slice of an A100 80GB. However, once the training data for the semantic entropy probes has been created, a CPU-only computing resource is sufficient to fit the logistic regression models.

Based on tracked finished runs, we estimate ~300 GPU-hours plus ~310 CPU-hours to obtain the results in the paper.

