# OpenReview forum: "Semantic Entropy Probes: Robust and Cheap Hallucination Detection in LLMs"
_ICLR.cc/2025/Conference — Submitted to ICLR 2025_

### Official Review · Reviewer_eRJ8 · 2024-11-02

**Soundness:** 3
**Presentation:** 3
**Contribution:** 2
**Rating:** 5
**Confidence:** 4

**Summary:**

This paper proposes Semantic Entropy Probes (SEPs) for hallucination detection in LLMs. The primary motivation of this work is to reduce the computational cost associated with semantic entropy, a representative sampling-based method for detecting hallucinations. SEPs are trained using the hidden states of LLMs to predict semantic entropy. The authors demonstrate that SEPs can effectively predict hallucinations and offer better generalization compared to probes that are directly trained for accuracy.

**Strengths:**

1. The motivation is intuitive and clear, the paper is well written and easy to implement.
2. Training linear probes on the hidden states of large language models to capture semantic entropy is a promising approach. It offers new insights for reducing the cost of traditional sampling-based hallucination detection.
3. Comprehensive ablation studies have been conducted across various models, tasks, layers, and token positions. The experimental results align with expectations.

**Weaknesses:**

1. The related work section lacks several crucial references [1,2,3,4]. For example, ref [1]  also adopt a supervised method for uncertainty estimation of LLMs. ref [2,3] explores the internal states of LLMs for hallucination detection. The author should introduce more hallucination detection methods based on internal states or linear probs. I have seen at least 5 papers that train a classifier for hallucination detection. Therefore, I suggest the authors to add a paragraph discussing internal state-based and probe-based hallucination detection methods, highlighting key differences between those approaches and SEPs

2. The performance of SEPs is unsatisfactory, and the author should include more competitive baseline methods. According to Fig. 7, SEPs fall short by 5-10 points compared to SE. Furthermore, SE is not the state-of-the-art (SOTA) method for hallucination detection [2,5]. It is recommended that the author include comparisons with the SOTA methods. I acknowledge that SEPs aim to balance performance and computational efficiency, I would suggest the authors to discuss this trade-off more explicitly in relation to SOTA methods.

3. There seems to be no clear explanation for why predicting SE yields better performance compared to predicting labels. I would appreciate a more detailed analysis explaining why predicting SE outperforms label prediction.

[1] Uncertainty Estimation and Quantification for LLMs: A Simple Supervised Approach

[2] INSIDE: LLMs' Internal States Retain the Power of Hallucination Detection

[3] LLM Internal States Reveal Hallucination Risk Faced With a Query.

[4] LLM Factoscope: Uncovering LLMs’ Factual Discernment through Measuring Inner States

[5] SelfCheckGPT: Zero-Resource Black-Box Hallucination Detection for Generative Large Language Models

**Questions:**

1. α represents the optimal threshold for partitioning raw SE scores into high categories, and I'm interested in how this threshold varies across different datasets. Additionally, I'm uncertain about the generalizability of trained SEPs across these datasets.
2. Would using hidden state information to predict SelfCheckGPT scores [1], Eigenscore [2], or other uncertainty scores potentially lead to better results?

[1] SelfCheckGPT: Zero-Resource Black-Box Hallucination Detection for Generative Large Language Models

[2] INSIDE: LLMs' Internal States Retain the Power of Hallucination Detection

---

> ### Author Response · Authors · 2024-11-20
> **Author Response to Reviewer eRJ8 (Part 1)**
>
> Dear reviewer eRJ8,
>
> Thank you for your hard work and helpful feedback. We have gladly incorporated many of your excellent suggestions into the draft and look forward to engaging with you during the discussion period to clarify any remaining points. Please read our comment above, addressed to all reviewers, first.
>
> > The related work section lacks several crucial references [1,2,3,4]. For example, ref [1] also adopt a supervised method for uncertainty estimation of LLMs. ref [2,3] explores the internal states of LLMs for hallucination detection. [...] Therefore, I suggest the authors to add a paragraph discussing internal state-based and probe-based hallucination detection methods, highlighting key differences between those approaches and SEPs
>
> Thank you for suggesting these related papers. We have updated the draft to add a discussion of them in relation to our work, and have added a slightly more extensive discussion below for your convenience.
>
> In [1], the authors train probes supervised by accuracy metrics, and we will cite them as another example of an accuracy-based probing method. They use random forests instead of linear models. We have investigated non-linear probing methods in Appendix A, but find no benefit from them. Further, the paper sometimes uses a different model for probing than is used to generate the answer, which we have certain objections with. Concerningly, this seems to rely entirely on knowledge being highly correlated between different models. This has also been highlighted in concurrent [reviews](https://openreview.net/forum?id=g3aGMMFHW0) of this work.
>
> In [2], the authors predict hallucinations by computing a so-called eigenscore from the latent states of _multiple_ model generations.  In other words, like full semantic entropy, [2] requires the sampling of multiple model generations. Therefore, this method does not call for direct comparison to SEPs given it incurs much higher computational costs. (It would be interesting to compare this method to SE though, and then perhaps try to approximate [2] from single generations similar to our work here.)
>
> [3] is another example of an accuracy-based probing method and we will cite them as such. They also present interesting experiments on predicting training set membership from latent states. We don’t think their probing method contains any methodological changes that would require us adding to/changing our accuracy probing baseline.
>
> [4] targets word-level factuality. It is unclear how to apply their method to the task of assigning overall factuality scores in the open-ended question-answering settings that we study in this paper. However, we think it would be interesting for future work to extend SEPs to word-level factuality and then to compare against [4].
>
> **Please read the second part of our rebuttal to your review next.**

---

> ### Author Response · Authors · 2024-11-20
> **Author Response to Reviewer eRJ8 (Part 2)**
>
> **Please start by reading the first part of our rebuttal to your review.**
>
> > The performance of SEPs is unsatisfactory, and the author should include more competitive baseline methods. According to Fig. 7, SEPs fall short by 5-10 points compared to SE. Furthermore, SE is not the state-of-the-art (SOTA) method for hallucination detection [2,5]. It is recommended that the author include comparisons with the SOTA methods. I acknowledge that SEPs aim to balance performance and computational efficiency, I would suggest the authors to discuss this trade-off more explicitly in relation to SOTA methods.
>
> Thank you for acknowledging that SEPs present a ‘best-in-class’ performance given their low computational cost. We would like to again emphasize the dramatic difference between SEPs and SE in terms of their test-time cost to quantify uncertainty. The cost of computing SEPs is basically negligible: some back-of-the-envelope math shows that SEPs add only about 0.0000002% in terms of FLOPS compared to the cost of an average generation for Llama-2-70B. This is compared to full SE, which increases test-time generation costs by a factor of K, with K=10 typical, and further require significant computational overhead to run the semantic clustering algorithm.
>
> We are further unsure how [2,5] show that SE is not a SOTA method. [5] targets long paragraph-length generations with multiple facts and focuses on how to decompose uncertainty across facts. As such, it does not compare to SE. Further [2],  also does not compare to SE. Also note that [2] seems to only study very short generations, see their examples in Appendix H, which is significantly less complex than the full sentence-length generations that we study following Farquhar et al.
>
> We would agree that the field of hallucination detection is currently lacking standardized definitions, experiment scenarios, and comprehensive benchmarks of methods. To us, it is unclear which of these methods is best at extracting model uncertainty! We think this is a problem in the field, and we would welcome future work that clearly compares all of the existing methods. We further think that using other types of uncertainty metrics as probing targets is exciting future work. In Appendix A (Figure A.11), we already present some interesting results when using the p(True) uncertainty method as a supervisory signal.
>
> Note that we already clearly state that SEPs do not match the performance of full SE in lines 259 and 465.  However, following your feedback, we have added a limitations paragraph to section 8, where we make clear again that SEPs cannot match the performance of full SE.
>
> **Please read the third part of our rebuttal to your review next.**

---

> ### Author Response · Authors · 2024-11-20
> **Author Response to Reviewer eRJ8 (Part 3)**
>
> **Please start by reading the first two parts of our rebuttal to your review.**
>
> > There seems to be no clear explanation for why predicting SE yields better performance compared to predicting labels. I would appreciate a more detailed analysis explaining why predicting SE outperforms label prediction.
>
> We agree this is a really interesting question that we could have spent more time on in the paper. Following your feedback we have reworked and improved our discussion in Section 8 of why SEPs generalize better to new tasks than accuracy probes and present a more extensive version of this for your convenience below.
>
> Our intuition here is that accuracy probes are more likely to latch on to spurious correlations between model predictions and accuracy in the dataset that will not generalize to new datasets, whereas SEPs will learn to robustly predict SE, which predicts accuracy well across tasks.
>
> More concretely, we believe that largely unavoidable problems with assigning accuracy labels to ultimately vague linguistic statements are to blame for this behavior.  While semantic uncertainty should generally be a good predictor of model accuracy, there may be some cases where uncertainty is _low_ despite the model answer being judged as _incorrect_. (This mostly happens due to shortcomings in measuring accuracy, e.g. the model answer not being precise enough, for example, just giving the year when the gold answer is looking for an exact date.) We believe these cases may lead to accuracy probes learning worse predictors than SE probes, and that this becomes noticeable on OOD data.
>
> In this scenario, for the SE probe, everything works according to plan: the label will be ‘high semantic uncertainty’, which we assume can be robustly predicted from latent space as we have discussed above. So if the probe is trying to find the direction in latent space that is predictive of ‘high semantic uncertainty’, the information it is getting with this datapoint agrees with previous observations. We further know that SE works robustly across datasets, so if the probe finds a direction in latent space that robustly represents SE, it will generalize well.
>
> For the accuracy probe, however, this datapoint is much more problematic. For it, the ‘SE uncertainty signal’ in latent space is not helpful for predicting the accuracy label anymore. Therefore, the accuracy probe is much less likely to learn to rely on the ‘SE uncertainty’ direction in latent space. Instead then, we believe that the accuracy probe likely relies on other features of the question embedding to achieve good predictive performance. Given the complexity of the hidden space, the accuracy probe may rely on features that identify questions the model cannot answer (e.g. by their topic, such as ‘historical facts’) instead of using the SE direction. However, these features would often be brittle and not generalize well to new datasets, as they conflate model capability with the labeling process. For example, we often seen models answering at the wrong level of granularity or missing implicit assumptions about for a particular subset of questions. For the OOD dataset, answer granularity and implicit assumptions may be different, such that these features will not generalize!
>
> (For example, on dataset A, the accuracy probe may learn that the LLM cannot answer historical questions, because the LLM always responds only with a year, when the gold answer is a full date. The accuracy probe will then learn to associate ‘historical’ questions with low accuracy – it will not rely on the SE signal as the model is confident in its answers here. However, on dataset B, the gold answer is only years and not full dates on similar historical questions. This will lead to failure of the accuracy probe. However, SEPs will work well on dataset B as they are much more likely to capture the underlying SE signal.)
>
> In summary, our results support the idea that semantic uncertainty can be captured robustly from latent space, with SEPs generalizing similarly to SE. Accuracy on the other hand depends on an ‘external’ labeling process, which the hidden states do not capture. Therefore, probe learning from accuracy signals is subject to conflicting signals and spurious correlations, which lead to worse performance in generalization regimes.
>
> **Please read the fourth part of our rebuttal to your review next.**

---

> ### Author Response · Authors · 2024-11-20
> **Author Response to Reviewer eRJ8 (Part 4)**
>
> **Please start by reading the first three parts of our rebuttal to your review.**
>
> > α represents the optimal threshold for partitioning raw SE scores into high categories, and I'm interested in how this threshold varies across different datasets.
>
> Following your question, we have updated the draft to show the best-split objective for Llama-2-7B across tasks in Appendix A (‘Different Binarization Procedures’, Figure A.15). We find that the objective behaves quite similarly across tasks. This complements our previous results in Figure A.14 which show the behavior of the global (across tasks) best-split objective for all models. Note that, for all our experiments, we rely on a single global threshold per model that is fixed across tasks.
>
> > Additionally, I'm uncertain about the generalizability of trained SEPs across these datasets.
>
> Could you perhaps clarify what we can do to alleviate your concerns about the generalizability? We feel that Figures 7 and 8, as well as Table 2, show clearly that SEPs generalize significantly better to new tasks than accuracy probes.
>
> > Would using hidden state information to predict SelfCheckGPT scores [1], Eigenscore [2], or other uncertainty scores potentially lead to better results?
>
> We think that using other uncertainty metrics, such as those proposed in [1] or [2], to supervise linear probes is a very interesting direction for future research. On a more abstract level, we suspect that LLM latent spaces generally capture uncertainty over semantic meaning. We do not know which uncertainty method, be it SE, [1], [2], or something else, best captures this underlying ‘ground-truth’ semantic uncertainty. Therefore, we think it is highly interesting future work to supervise linear probes with other uncertainty methods. In fact, in Appendix A (Figure A.11), we already present some interesting preliminary results when using the p(True) uncertainty method as a supervisory signal.
>
> Again, thank you very much for your review! Please let us know if you have any remaining questions or concerns and we will do our best to respond to them as soon as possible.

---

> > ### Comment · Reviewer_eRJ8 · 2024-11-25
> >
> > Thank you for providing a detailed response along with additional experiments and explanations in the revised version. Most of my concerns have been addressed. However, I'm still not fully convinced about SEP's performance and the explanation of why SEP is considered superior to Accuracy Prob.
> >
> > - In Figure A.8-A.10, SEP underperforms Accuracy Prob on almost half of the models or evaluation datasets.
> > The observation that SEP demonstrates superior OOD generalization performance raises concerns about the potential inadequate training of Accuracy Prob. Would incorporating a regularization term into the training process help prevent overfitting and improve the generalization performance of Accuracy Prob?
> >
> > - Furthermore, the discussion section offers only subjective conjectures from the authors regarding why SEPs should perform better than Accuracy Prob in OOD settings. The absence of theoretical insights or experimental support makes it difficult for readers to be fully convinced of why predicting SE could have greater potential than predicting labels.
> >
> > I maintain my negative score because the authors have not fully convinced me of the superiority of SEPs over Accuracy Prob.

---

> > > ### Author Response · Authors · 2024-11-25
> > > **Author Response**
> > >
> > > Thank you for taking the time to read our rebuttal! We are glad we could address most of your concerns and hope to clear up any that remain.
> > >
> > > > In Figure A.8-A.10, SEP underperforms Accuracy Prob on almost half of the models or evaluation datasets. The observation that SEP demonstrates superior OOD generalization performance raises concerns about the potential inadequate training of Accuracy Prob. Would incorporating a regularization term into the training process help prevent overfitting and improve the generalization performance of Accuracy Prob?
> > >
> > > We are already using L2 regularization (see Section 5, 'Linear Probe' ) and find that it is helpful for the performance of both probes. We have tried other regularization techniques in preliminary experiments but have found L2 regularization to work best for both probes. We have taken every care to ensure that the performance of the accuracy probe is as good as it can be and are happy to try our best to perform any additional experiments that you may wish to see in this direction!
> > >
> > > Further, we do not think that the worse generalization performance of accuracy probes is a sign of inadequate regularization during training. Even in-distribution, all accuracies that we report are computed on a separate test set that the probes are not trained on. If the regularization of the accuracy probe was inadequate, then we would expect the test set performance to be bad in-distribution already, right?
> > >
> > > Instead, we believe our experiments conclusively show that SEPs learn _different_ features than accuracy probes and thus generalize better than them to new tasks.
> > >
> > > > Furthermore, the discussion section offers only subjective conjectures from the authors regarding why SEPs should perform better than Accuracy Prob in OOD settings. The absence of theoretical insights or experimental support makes it difficult for readers to be fully convinced of why predicting SE could have greater potential than predicting labels.
> > >
> > > We are excited for future work to further concretize our explanation for why SEPs perform better than accuracy probes out-of-distribution. If you think that this section is misleading about its conjectural nature at any point, please let us know and we will improve our wording.
> > >
> > > However, we disagree strongly that there is no experimental support for our claims that SEPs perform better out-of-distribution than accuracy probes. Table 2 shows that SEPs perform significantly better than accuracy probes (at 2 std error level) for all models that we consider. Further, this is not a property that appears on average over tasks only: Figures 7 and 8 show SEPs generalize better for all models and tasks individually as well.
> > >
> > > Again, thank you for interacting with us during the discussion period! Please get back to us if we have been unable to clear up your remaining concerns.

---

> > > > ### Author Response · Authors · 2024-12-02
> > > > **Author Reminder**
> > > >
> > > > Dear reviewer eRJ8,
> > > >
> > > > Tomorrow is the last day for authors to post messages on the forum. We are glad to hear that we could already address most of your concerns with our initial rebuttal, and we hope have addressed all remaining concerns in our follow-up message above.  We would be grateful if this was sufficient for you to increase your score to an accept rating. If there is anything else we can clarify or do, please let us know as soon as possible.
> > > >
> > > > Many thanks in advance,
> > > > The Authors

---

### Official Review · Reviewer_BxzX · 2024-11-03

**Soundness:** 3
**Presentation:** 3
**Contribution:** 3
**Rating:** 6
**Confidence:** 3

**Summary:**

The paper describes an approach to hallucination detection. Unlike existing methods, that generate multiple samples at inference time to derive an estimate of uncertainty (semantic entropy), the method trains a linear model on a subset of the LLM's hiddens to predict it efficiently. The paper shows that this works reasonably well, although the accuracy is behind that of the sampling based methods.

**Strengths:**

The paper is well-written, quite polished and very easy to follow. Empirical evaluations seem reasonably detailed and the findings clear.

**Weaknesses:**

The accuracy of the semantic entropy prediction is quite significantly behind the sampling-based estimate. In addition, it seems hard to quantify how much prediction accuracy for hallucination detection would actually impact any practical applications. Since there are many possible avenues to improve accuracy, some investigation in this direction - even just in a preliminary fashion - would make this work stronger.

**Questions:**

The improved efficiency comes at a significant accuracy hit as compared to SE. How would the SE results be affected for smaller sample size N? The potential benefits of the proposed method would be significantly diminished if SE tends to be more or less stable in response to reducing N.

Would it make sense to use significantly higher N during training as a way to improve SEP accuracy?

In this work, probes are applied only to separate layers. Have the authors considered combinations of multiple layers? This would make sense if one assumes that different layers may contribute different evidence for model uncertainty.

How OOD is the OOD setting really? How (dis-)similar are the different datasets from one another?

I assume the values in Figure 2 are for held-out not training data. Is that correct?

What is the actual potential time efficiency benefit of the method as compared to SE, for example, as measured in in wall clock time? And how much would the possibility to generate samples in batch-mode affect those numbers?

The proposed method is very simple and seems somewhat "obvious". I'm a bit surprised this hasn't been studied before, but not being an expert in this work I give the paper the benefit of the doubt and consider this simplicity a plus.

---

> ### Author Response · Authors · 2024-11-20
> **Author Response to Reviewer BxzX (Part 1)**
>
> Dear reviewer BxzX,
>
> Thank you for your hard work and helpful feedback. We have gladly incorporated many of your excellent suggestions into the draft and look forward to engaging with you during the discussion period to clarify any remaining points. Please read our comment above, addressed to all reviewers, first.
>
> > The accuracy of the semantic entropy prediction is quite significantly behind the sampling-based estimate.
>
> We agree that SEPs’ performance lags behind SE and other sampling-based approaches. We believe that closing this gap is exciting future work. For now, SEPs offer a “best-in-class” performance considering their extremely low computational cost.
>
> > In addition, it seems hard to quantify how much prediction accuracy for hallucination detection would actually impact any practical applications. Since there are many possible avenues to improve accuracy, some investigation in this direction - even just in a preliminary fashion - would make this work stronger.
>
> This is a great suggestion! We have added accuracy rejection curves to Appendix A (Figure A.16). These curves show that SEPs can improve the overall accuracy of the model by refusing to predict on those inputs where SEPs predict the highest uncertainty. This is only possible because high a high predictive value from SEPs correlates well with model accuracy, such that low uncertainty inputs (as judged by SEPs) are less likely to be correct. We believe that such rejection accuracy curves are a great way to showcase the positive practical impact of SEPs and plan to add a more extended showcase for the camera ready version of the paper.
>
> > Would it make sense to use significantly higher N during training as a way to improve SEP accuracy?
>
> We here follow Farquhar et al., who typically set N=10, which robustly gives good performance across tasks in their experiments. They also provide an ablation which shows that SE performance does not significantly improve for N larger than 10. As SEPs use only the SE value as a supervisory signal, we don’t expect them to benefit from significantly larger N, as such increases in N don’t benefit SE itself either.
>
> > In this work, probes are applied only to separate layers. Have the authors considered combinations of multiple layers? This would make sense if one assumes that different layers may contribute different evidence for model uncertainty.
>
> Yes, we have. In fact, some of the results in the current version of the draft do concatenate hidden states across layers for probe training. We detail this in Appendix B.4 (‘layer concatenation’) and have revised the main version of the draft to point more clearly to the appendix here. We generally find that layer concatenation helps moderately improve performance. Following your feedback we have added a direct comparison of results with and without layer concatenation in Figure B.3.
>
> > How OOD is the OOD setting really? How (dis-)similar are the different datasets from one another?
>
> That’s a great question! All the datasets we consider are free-form question-answering datasets. TriviaQA has trivia questions, NQOpen real-world questions people asked Google, SQuAD is a reading comprehension task (but its questions can usually be answered without the context), and BioASQ targets medical and biological domain knowledge. Although this is difficult to quantify, we would say that, from a human perspective, TriviaQA, NQOpen, and SQuAD are all quite similar in terms of the type of questions that they contain and that BioASQ is the most unique. In a sense, it is therefore even more surprising that accuracy probes can struggle significantly to transfer between these datasets.
>
> > I assume the values in Figure 2 are for held-out not training data. Is that correct?
>
> Correct! This is data that was not used to train the probes in any way.
>
> **Please read the second part of our rebuttal to your review next.**

---

> ### Author Response · Authors · 2024-11-20
> **Author Response to Reviewer BxzX (Part 2)**
>
> **Please start by reading the first part of our rebuttal to your review.**
>
> > What is the actual potential time efficiency benefit of the method as compared to SE, for example, as measured in in wall clock time? And how much would the possibility to generate samples in batch-mode affect those numbers?
>
> Following your feedback, we have collected the actual time difference between computing SE and SEPs for Llama-2-7B on TriviaQA. We find that, on average, computing SE for 1,000 samples takes 168 $\pm$ 19.11 minutes. In contrast, the computation of SEP predictions requires only 7.13 $\pm$ 0.14 seconds in the same setting. Consequently, SEPs are able to speed up hallucination detection by 1413-fold compared to full SE. We have added these results to Appendix A.
>
> Note that, in terms of pure FLOPS, SEPs are much more than 1400 times cheaper than SE. Some back-of-the envelope math suggests that SEPs add only about 0.0000002% in terms of FLOPS over naive generation, suggesting our current wall-clock speedup could be improved significantly.
>
> As for parallelizing over batches, SEPs are just linear layers and so they can be computed in parallel without any problem. For SE, while the additional generations can be performed in parallel, the semantic clustering algorithm and NLI computations are run sequentially (iteratively over the set of generations for each question).
>
> > The proposed method is very simple and seems somewhat "obvious". I'm a bit surprised this hasn't been studied before, but not being an expert in this work I give the paper the benefit of the doubt and consider this simplicity a plus.
>
> Great! We agree that the method is simple and would also consider this a strength. We also think that some of the ideas around SEPs _generalizing_ better than accuracy probes are perhaps less obvious and may explain why SEPs have not been considered before.
>
> Again, thank you very much for your review! Please let us know if you have any remaining questions or concerns and we will do our best to respond to them as soon as possible.

---

> > ### Comment · Reviewer_BxzX · 2024-11-25
> > **Follow-up on remaining open question**
> >
> > Thank you for your comments so far. Could you also comment on the following open question in the review:
> > "The improved efficiency comes at a significant accuracy hit as compared to SE. How would the SE results be affected for smaller sample size N? The potential benefits of the proposed method would be significantly diminished if SE tends to be more or less stable in response to reducing N."
> >
> > Just to avoid any potential confusion, please note that this question is about reducing N when using SE. It is not about the unrelated question on increasing N for training SEP, which was already addressed in the rebuttal.

---

> > > ### Author Response · Authors · 2024-11-26
> > > **Author Response**
> > >
> > > Dear reviewer BxzX,
> > >
> > > Thank you for bringing this to our attention and apologies for failing to directly respond to this question of yours initially.
> > >
> > > > The improved efficiency comes at a significant accuracy hit as compared to SE. How would the SE results be affected for smaller sample size N? The potential benefits of the proposed method would be significantly diminished if SE tends to be more or less stable in response to reducing N.
> > >
> > > The performance of SE suffers significantly as the number of samples is reduced, see Figure 3 (a) in Kuhn et al. (https://arxiv.org/pdf/2302.09664).
> > >
> > > Further note that the lowest N=1 reported by Kuhn et al. corresponds to one additional high temperature generation. Even in this scenario, the cost of computing SEPs is _drastically_ lower than the cost of computing SE (with only a single additional generation). The back-of-the envelope math we have mentioned in our rebuttal states that the cost of SEPs compared to a model generation is 0.0000002% in terms of FLOPS. So even in the cheapest-case scenario, when computing SE with only a single additional generation, this would mean that SEPs still cost less than 0.0000002% of the cost of computing SE.
> > >
> > > Again, apologies for missing this initially and please let us know if you have any remaining questions or concerns that stand in the way of a potential increase in your score!

---

> > > > ### Author Response · Authors · 2024-12-02
> > > > **Author Reminder**
> > > >
> > > > Dear reviewer BxzX,
> > > >
> > > > Tomorrow is the last day for authors to post messages on the forum. We hope our rebuttal has clarified any remaining concerns on your side. If there is anything we can do to make it easier for you to support the acceptance of our submission, or to increase your rating, please let us know as soon as possible!
> > > >
> > > > Many thanks in advance,
> > > > The Authors

---

### Official Review · Reviewer_toQ8 · 2024-11-04

**Soundness:** 3
**Presentation:** 3
**Contribution:** 2
**Rating:** 5
**Confidence:** 4

**Summary:**

The paper proposes a new method Semantic Entropy Probes (SEPs) for hallucination detection of LLMs, which aims to capture semantic entropy by training a linear classifier on top of LLM hidden states. The authors perform various ablation studies to understand different design choices (model family, layer, token position) across common benchmarks. Experiments suggest the effectiveness of the proposed method over various baselines.

**Strengths:**

- The method to estimate semantic entropy by linear probing is simple to implement and alleviates repeated sampling at test-time.
- The experiments and ablations are thorough to demonstrate the effectiveness of the proposed method.
- The overall organization of the paper is clear and coherent.

**Weaknesses:**

- **Motivation of SEP Efficiency**: The primary motivation for SEP, as stated in the paper, is that it is "cheap" and significantly enhances computational efficiency compared to the baseline semantic entropy (SE). However, this claim appears to be poorly substantiated. The comparison between a training-based method (SEP) and a test-time method (SE) is insufficiently justified. For a fair comparison, it might be better to provide analysis of the overall cost: SE with sampling v.s. the total cost (training and inference) of SEP.

- **Improvement in Long Generation Experiments**: The long generation experiments could be improved. Currently, long generation is primarily conducted with prompts that request "a single brief but complete sentence," resulting in relatively short responses. This approach diverges from recent practices, where models are often prompted to generate detailed reasoning or more elaborate outputs for long-form generation.

**Questions:**

- To calculate semantic entropy (L185), how is the number of clusters K determined?
- Can authors further discuss why "the discrete variant of SE" should be used instead of the plain version Eq (1)? How is the "fraction of generations in the cluster" related to "avoid problems for generation of different length" (L193)?
- In Sec 5, it seems that the prompt "single brief but complete sentence" also promotes short responses. Does a Chain-of-Thought style prompt work for the long-form answer generation?

---

> ### Author Response · Authors · 2024-11-20
> **Author Response to Reviewer toQ8 (Part 1)**
>
> Dear reviewer toQ8,
>
> Thank you for your hard work and helpful feedback. We have gladly incorporated many of your excellent suggestions into the draft and look forward to engaging with you during the discussion period to clarify any remaining points. Please read our comment above, addressed to all reviewers, first.
>
> > Motivation of SEP efficiency: [...]
>
> Thanks for bringing this to our attention! It is absolutely true that SEPs need to be trained while SE can just be calculated directly at test time without any training and, further, the initial training of SEPs does require SE computation for many training sequences. However, we do believe that our claims of SEPs lower computational costs at test time can be extended to apply to the _overall_ cost, including SEP training cost, in many practical settings.
>
> For one, the cost of computing SEPs at test time is basically negligible: some back-of-the-envelope math (we’re happy to add this) shows that SEPs add only about 0.0000002% in terms of FLOPS compared to the cost of an average generation for Llama-2-70B. This is compared to full SE, which increases test-time generation costs by a factor of K, with K=10 typical, and further requires significant computational overhead to run the semantic clustering algorithm.
>
> So for the comparison that you suggest, one would essentially find that SEPs become cheaper than SE as soon as the size of the test queries is larger than the size of the training set. We use training sets with between 1000 and 2000 samples, which we believe is a number that should quickly be reached in large-scale deployment scenarios of LLMs.
>
> Following your feedback, we have added a new discussion of limitations in Section 8, where we clearly state that SEPs are only compute-efficient overall if the number of training samples is much smaller than the number of test queries. (Which again, we believe should be quite a reasonable assumption in most cases.)
>
> > Improvement in Long Generation Experiments: [...]
>
> Thanks for raising this concern! With our ‘short’ and ‘long’ generation scenarios, we directly follow Farquhar et al. (2024) in both notation and experimental setting.
>
> We agree that extending our work to _paragraphs_ of generations, i.e. multi-sentence response that contain a number of factual statements, is exciting work. However, we note that, as far as we are aware of, most approaches for detecting hallucinations in paragraphs boil down to first extracting _individual_ factual statements and then testing each factual statement on its own (see e.g., Farquhar et al., 2024, arXiv:2303.08896, arXiv:2305.15852, arXiv:2309.11495, 2210.08726). As such, we believe that research into methods such as ours, which improve the factuality detection of individual statements, continues to be valuable, as they can be combined with many of the paragraph-length methods to serve as the core hallucination detector.
>
> Further, a lot of other recent approaches for hallucination detection also focus on the setting of detecting hallucinations where model responses are as long or shorter than in our publication, see for example arXiv:2404.15993, arXiv:2402.03744, arXiv:2407.03282, arXiv:2307.01379, arXiv:2305.14613, arXiv:2302.09664. There is therefore clearly an interest in the community for detecting hallucinations in this setting. Note that our long generation prompt leads to model answers with an average length of 96 characters, which we suspect is much longer than many of the above citations.
>
> Following your feedback, we have further investigated chain-of-thought prompting for our setup (see below).
>
> > To calculate semantic entropy (L185), how is the number of clusters K determined?
>
> We here rely on the clustering algorithm proposed by [Farquhar et al.](https://www.nature.com/articles/s41586-024-07421-0), see their Methods section and Extended Data Fig. 1. In this algorithm, the number of clusters K is not fixed for a given question. Instead K will correspond to the number of distinct semantic meanings found by the algorithm when given N sampled model generations.
>
> Intuitively, the algorithm proceeds as follows. For each model generation, compare the generation to all existing clusters. If the generation has the same semantic meaning as one of the existing clusters, add it to that cluster. Else, create a new cluster containing that generation.
>
> We have updated the draft to clarify that K is not fixed and corresponds to the number of distinct semantic clusters found by the algorithm.
>
> **Please read the second part of our rebuttal to your review next.**

---

> ### Author Response · Authors · 2024-11-20
> **Author Response to Reviewer toQ8 (Part 2)**
>
> **Please start by reading the first part of our rebuttal to your review.**
>
> > Can authors further discuss why "the discrete variant of SE" should be used instead of the plain version Eq (1)? How is the "fraction of generations in the cluster" related to "avoid problems for generation of different length" (L193)?
>
> Both the discrete and plain version of SE could be used with SEPs. We simply prefer the discrete version following the recent publication of Farquhar et al., who find it yields comparable performance while being simpler conceptually and allowing application to models where tokens probabilities are not accessible.
>
> The discrete SE variant uses a simple counting measure to estimate the probability of a particular semantic meaning (e.g. the probability of semantic meaning A is given by the fraction of generations the clustering algorithm associates with that meaning). It does not use the token probabilities of the generated sequences.
>
> This is in contrast to the original SE variant introduced by Kuhn et al. which does rely on the token probabilities to estimate SE. A problem that comes up when doing this naively is that longer sequences will typically have lower probability because the probability of a sequence is a product of the conditional probabilities (<=1) of the individual tokens. Empirically, what works well is to compute the _average_ of the log token probabilities instead of their sum, i.e. divide the sum of log probabilities by the length of the generation (see citations at the end of Section 3). However, this is wrong from a probabilistic perspective as we are no longer working with the true joint log probability of the sequence. The discrete SE variant allows us to estimate semantic cluster probabilities without using token sequence probabilities. Thus we can avoid making a difficult choice regarding length-normalization of log sequence probabilities.
>
> > In Sec 5, it seems that the prompt "single brief but complete sentence" also promotes short responses. Does a Chain-of-Thought style prompt work for the long-form answer generation?
>
> Thank you for this suggestion! Following your feedback, we have prompted Llama-3 70B to ‘think step by step’ when giving answers on TriviaQA. Firstly, we have found that CoT prompting does not significantly change accuracy on this task ($0.885$ before vs $0.888$ after). Interestingly, however, the in-distribution performance of SEPs actually seems to improve with CoT prompting, whereas the performance of accuracy probes degrades to such a degree that SEPs seem to outperform accuracy probes even in-distribution.
>
> |          | Acc. Probes   | SEP                   |
> |----------|-------------------|-----------------------|
> |w/o CoT   |$0.835$  | $0.749$     |
> |with CoT  |$0.761$  | $0.788$     |
>
> We think these results are very interesting and plan to investigate CoT prompting further for the camera ready version of the paper.
>
>
> Again, thank you very much for your review! Please let us know if you have any remaining questions or concerns and we will do our best to respond to them as soon as possible.

---

> > ### Comment · Reviewer_toQ8 · 2024-12-02
> >
> > I appreciate the authors' detailed responses, which mostly addressed my concerns.
> >
> > - While the proposed SEP method demonstrates advantages over the test-time baseline SE, its applicability appears to hinge on the availability of sufficient training samples ("1000 and 2000 samples are required"). This requirement may pose challenges in many real-world applications with private data from customers, where only a limited number of samples are available. However, this trade-off is understandable, as there are other scenarios where ample training data is accessible.
> >
> > - The claims regarding "long generation" could be arguable depending on the literature (e.g., if "avg len of 96 characters" is considered reasonably long). More investigations are deferred to future work.
> >
> > Overall, I think the paper falls into the borderline, given both its contributions and limitations.

---

> > > ### Author Response · Authors · 2024-12-02
> > > **Author Response**
> > >
> > > Thank you for taking the time to read our rebuttal. We are glad to hear that we could mostly address your concerns and would be grateful if this was sufficient for you to increase your score to an accept rating.
> > >
> > > > While the proposed SEP method demonstrates advantages over the test-time baseline SE, its applicability appears to hinge on the availability of sufficient training samples ("1000 and 2000 samples are required"). This requirement may pose challenges in many real-world applications with private data from customers, where only a limited number of samples are available. However, this trade-off is understandable, as there are other scenarios where ample training data is accessible.
> > >
> > > We agree that this is a trade-off and that there will be many situations where SEPs training requirements are met. However, even if this is not the case, our results show that SEPs generalize much better to new tasks than accuracy probes. If no or not enough target data is available, one could train SEPs on data from other (possibly related) tasks and expect them to generalize well to the target data. We're happy to add this discussion to the next version of the draft.
> > >
> > > > The claims regarding "long generation" could be arguable depending on the literature (e.g., if "avg len of 96 characters" is considered reasonably long).
> > >
> > > Again, our generations are longer or as long as many other recent works on hallucinations, e.g. arXiv:2404.15993, arXiv:2402.03744, arXiv:2407.03282, arXiv:2307.01379, arXiv:2305.14613, arXiv:2302.09664. However, we are happy to add a disclaimer to the limitations section of the next version of the draft.

---

### Official Review · Reviewer_6ThF · 2024-11-06

**Soundness:** 3
**Presentation:** 3
**Contribution:** 3
**Rating:** 6
**Confidence:** 3

**Summary:**

This paper introduces a technique called Semantic Entropy Probes (SEP) for detecting hallucinations in large language models (LLMs). SEP leverages the internal states of LLMs to predict the semantic entropy (SE) of their responses. By doing so, the method empirically achieves an accurate approximation of SE without the computational overhead of multi-sampling. Furthermore, SEP effectively identifies model hallucinations in both in-distribution and out-of-distribution settings.

**Strengths:**

* This paper addresses a timely and significant research problem in LLMs. Detecting hallucinations is crucial for ensuring the safety of LLM applications in real-world scenarios.
* The experiments are extensive and compelling. The authors conduct experiments on and report results for several popular open-source models.
* The paper is generally well-written and easy to follow.

**Weaknesses:**

* I think the main weakness of this work is the lack of clear motivation and explanation for the outcomes. The SEP method appears to simply switch the prediction goal of previous internal state detection methods from accuracy to semantic entropy (SE), which seems like a simple 'a+b' combination. Although the results are promising, I am still expecting more insights from the experimental results regarding the mechanisms of LLMs. For example, why is it possible to use internal states to predict SE? And why does switching the prediction goal from accuracy to SE result in better performance in out-of-distribution settings? A clearer explanation of these questions would further enhance the solidness of this work and our understanding of LLMs.
* An obvious limitation is the applicability of the proposed method to black-box LLMs. I recommend that the authors discuss this in the limitations section.
* Most of the experiments compare the proposed method with accuracy-based probing methods. However, there are insufficient details provided to describe these counterpart methods. It would be more helpful to provide a comprehensive introduction to these methods for readers who are unfamiliar with them.

**Questions:**

1. Do authors have a clear explanation of why it is feasible to predict semantic entropy (SE) from internal states and why SE could serve as a better supervisory signal than accuracy in out-of-distribution training settings?
2. One of the advantages of SEP is that it does not rely on ground truth labels from the dataset. However, as mentioned by the authors, there are also existing works [1,2] that predict the trustworthiness of LLMs from internal states using unsupervised methods. I am curious about how these methods would perform under the same settings as SEP.


---
[1] Discovering Latent Knowledge in Language Models Without Supervision
[2] Representation Engineering: A Top-Down Approach to AI Transparency

---

> ### Author Response · Authors · 2024-11-20
> **Author Response to Reviewer 6ThF (Part 1)**
>
> Dear reviewer 6ThF,
>
> Thank you for your hard work and helpful feedback. We have gladly incorporated many of your excellent suggestions into the draft and look forward to engaging with you during the discussion period to clarify any remaining points. Please read our comment above, addressed to all reviewers, first.
>
> > I think the main weakness of this work is the lack of clear motivation and explanation for the outcomes. The SEP method appears to simply switch the prediction goal of previous internal state detection methods from accuracy to semantic entropy (SE), which seems like a simple 'a+b' combination. [...] why is it possible to use internal states to predict SE? And why does switching the prediction goal from accuracy to SE result in better performance in out-of-distribution settings?
>
>
> We think these are really interesting questions! Fortunately, we can actually offer some intuitions and insights into why SE can be predicted from latent space and why SEPs would consequently generalize better than accuracy probes, and we are happy to detail these below. In fact, these intuitions were the reason we originally explored SEPs in the first place, and we see our results as supportive evidence of these intuitions.  However, we agree that further work could be done to fully answer these questions and would welcome any concrete suggestions that you may have in this direction.
>
> **Why can we predict SE from latent space?** For the general-purpose QA setting that we study here, we think the following toy example provides helpful intuition to answer this question. Let’s say we ask the LLM “When was the Hoover Dam built?”, and let’s further assume the LLM does not know the answer to this question. In our experience, which is supported by Farquhar et al. and others, this will lead to high semantic uncertainty in the answers. Concretely, when sampling from the model, we will obtain answers such as “It was built in 1930”, “1931”, “In 1932”, “1930”, “The hoover damn was built in 1930”,  “In 1932”, or “1932”. The sampled answers vary in both syntax and semantic meaning. In terms of syntax, we have the variants “It was built in X”, “X”, and “In X”, where X is a year. And in terms of semantic meanings, we observe the years “1930”, “1931”, and “1932”, indicating high semantic uncertainty.
>
> Now, when looking at only the distribution of p(next_token|”When was the Hoover Dam built?”), based on the above generations, we know that the model assigns probability mass to tokens such as [It, The, In, 1930, 1931, 1932] (depending on the exact tokenizer of course). Crucially here, models almost always consider giving extremely short answers (which are just directly the year in this case). From this, we can conclude that the latent state before answering contains knowledge of the semantic meaning of the generated answer as well as the semantic uncertainty over it, even if the model ends up generating a longer-form answer such as “The hoover dam was built in X”. Note that this idea is more general than it may seem at first, as there is rarely a question which cannot be answered in just a few tokens.
>
> So, in summary, the latent space here must contain semantic-level information about the answer. It would be highly likely for the model to learn an efficient representation that separates semantics and syntax, such that it can keep track of high level information across tokens. Further, for the probabilities of the model to be calibrated (which is something that should be promoted from training with a proper scoring rule such as negative log likelihood loss), this latent state should correctly model uncertainty over the semantic meaning. By inspecting the token distributions, one can see that, for any given syntactic instantiation (such as “It was built in X”), the token probabilities are dispersed when semantic uncertainty is high. It seems to us that the simplest way to achieve this for a given model is to represent semantic meaning _and_ uncertainty in the latent space separately from syntax. This is in line with work such as Belrose et al. (arXiv 2303.08112) and Pal et al. (arXiv 2311.04897) who show that hidden states are predictive of outputs multiple tokens ahead with high accuracy, as well as represent and control higher level functions of LLM behavior (see our literature review).
>
> **Please read the second part of our rebuttal to your review next.**

---

> ### Author Response · Authors · 2024-11-20
> **Author Response to Reviewer 6ThF (Part 2)**
>
> **Please start by reading the first part of our rebuttal to your review.**
>
> **Why will SEPs generalize better to OOD settings than accuray probes?** This is a very good question, too. Our intuition here is that accuracy probes are more likely to latch on to spurious correlations between model predictions and accuracy in the dataset that will not generalize to new datasets, whereas SEPs will learn to robustly capture a semantic entropy signal from the latent space which predicts factuality well _across_ tasks.
>
> More concretely, we believe that largely unavoidable problems with assigning accuracy labels to ultimately vague linguistic statements are to blame for the deficiencies of accuracy probes.  While semantic uncertainty should generally be a good predictor of model accuracy, there may be some cases where uncertainty is _low_ despite the model answer being judged as _incorrect_ (Farquhar et al., 2024). (This mostly happens due to shortcomings in measuring accuracy, e.g. the model answer not being precise enough, for example, just giving the year when the gold answer is looking for an exact date.) We believe these cases may lead to accuracy probes learning worse predictors than SE probes, and that this becomes noticeable on OOD data.
>
> In this scenario, for the SE probe, everything works according to plan: the label will be ‘high semantic uncertainty’, which we assume can be robustly predicted from latent space as we have discussed above. So if the probe is trying to find the direction in latent space that is predictive of ‘high semantic uncertainty’, the information it is getting with this datapoint agrees with previous observations. We further know that SE works robustly across datasets, so if the probe finds a direction in latent space that robustly represents SE, it will generalize well.
>
> For the accuracy probe, however, this datapoint is much more problematic. For it, the ‘SE uncertainty signal’ in latent space is not helpful for predicting the accuracy label anymore. Therefore, the accuracy probe is much less likely to learn to rely on the ‘SE uncertainty’ direction in latent space. Instead then, we believe that the accuracy probe likely relies on other features of the question embedding to achieve good predictive performance. Given the complexity of the hidden space, the accuracy probe may rely on features that identify questions the model cannot answer (e.g. by their topic, such as ‘historical facts’) instead of using the SE direction. However, these features would often be brittle and not generalize well to new datasets, as they conflate model capability with the labeling process. For example, we often seen models answering at the wrong level of granularity or missing implicit assumptions about for a particular subset of questions. For the OOD dataset, answer granularity and implicit assumptions may be different, such that these features will not generalize!
>
> (For example, on dataset A, the accuracy probe may learn that the LLM cannot answer historical questions, because the LLM always responds only with a year, when the gold answer is a full date. The accuracy probe will then learn to associate ‘historical’ questions with low accuracy – it will not rely on the SE signal as the model is confident in its answers here. However, on dataset B, the gold answer is only years and not full dates on similar historical questions. This will lead to failure of the accuracy probe. However, SEPs will work well on dataset B as they are much more likely to capture the underlying SE signal, which leads to better generalization.)
>
> In summary, our results support the idea that semantic uncertainty can be captured robustly from latent space, with SEPs generalizing similarly to SE. Accuracy on the other hand depends on an external labeling process, which the hidden states do not capture. Therefore, probe learning from accuracy signals is subject to conflicting information and spurious correlations, which lead to worse performance in generalization regimes.
>
> Again, thank you for raising both of these questions. Following your feedback, we have improved our discussion of this aspect of SEPs in Section 8 in the current version of the draft.
>
> > An obvious limitation is the applicability of the proposed method to black-box LLMs. I recommend that the authors discuss this in the limitations section.
>
> We agree that this is a limitation of SEPs (and probing methods in general), and have added a discussion of this to a new paragraph on limitations in Section 8.
>
> **Please read the last part of our rebuttal to your review next.**

---

> ### Author Response · Authors · 2024-11-20
> **Author Response to Reviewer 6ThF (Part 3)**
>
> **Please start by reading the first two parts of our rebuttal to your review.**
>
> > Most of the experiments compare the proposed method with accuracy-based probing methods. However, there are insufficient details provided to describe these counterpart methods. It would be more helpful to provide a comprehensive introduction to these methods for readers who are unfamiliar with them.
>
> Thank you for the suggestion! Following your feedback, we have expanded our discussion on accuracy probing in the related works section.
>
> > 1. Do authors have a clear [...]
>
> If we’re not mistaken, our answer to your first point above already addresses these questions.
>
> > One of the advantages of SEP is that it does not rely on ground truth labels from the dataset. However, as mentioned by the authors, there are also existing works [1,2] that predict the trustworthiness of LLMs from internal states using unsupervised methods. I am curious about how these methods would perform under the same settings as SEP.
>
> Thanks for bringing this to our attention! With regard to [1], their proposed CCS method can only be applied to binary yes/no questions if we understand correctly. This setting is more restrictive than the general question answering scenario that we discover here. There further seem to be some problems with the robustness of this method as discussed by Farquhar et al. (2312.10029). [2] does present a number of very interesting ideas but note that their method is actually not fully unsupervised as they rely on labels to ‘identify the layer and direction’ of the probes for truthfulness prediction (Section 4.1). We have added a discussion of these work to our related work section.
>
> Again, thank you very much for your review! Please let us know if you have any remaining questions or concerns and we will do our best to respond to them as soon as possible.

---

> > ### Author Response · Authors · 2024-12-02
> > **Author Reminder**
> >
> > Dear reviewer 6ThF,
> >
> > Tomorrow is the last day for authors to post messages on the forum. We hope our rebuttal has clarified any remaining concerns on your side. If there is anything we can do to make it easier for you to champion the acceptance of our submission, or to increase your rating, please let us know as soon as possible!
> >
> > Many thanks in advance,
> > The Authors

---

### Author Response · Authors · 2024-11-20
**Author Response to All Reviewers**

We thank all reviewers for their thoughtful reviews, hard work, and valuable suggestions, which we believe have already strengthened our submission. We were glad to see you all appreciate that our results demonstrate that semantic entropy probes provide a cheap and effective approximation of full semantic entropy, outperforming more conventional accuracy-based probing methods in terms of generalization performance.

We particularly appreciate the positive recognition of many aspects of our work, such as our experimental evaluations and ablations, which you have highlighted as ‘extensive and compelling’ (6ThF), ‘thorough’ (toQ8), ‘comprehensive’ (eRJ8), ‘reasonably detailed’ (BxzX) with ‘clear’ (BxZx) findings, and as ‘demonstrat[ing] the effectiveness of the proposed method’ (toQ8). We were further glad to see you felt our paper ‘addresses a timely and significant research problem’ (6ThF) with ‘intuitive and clear’ (eRJ8) motivation, positively highlighting the simplicity of our method and its implementation (6ThF, toQ8, eRJ8, BxzX), as well as recognizing our draft as 'well-written' (6ThF, BxzX, eRJ8), 'clear and coherent' (toQ8), and 'easy to follow' (6Thf, BxZX).

Following your excellent feedback, we have made a number of improvements to our submission. In particular:


* Add a demonstration of real-world computational savings when using SEPs over full SE computation. We observe a 1400 fold speedup in terms of computation time (Appendix A, ‘Computational Costs for SEPs and SE’).
* Add results comparing layer-concatenated SEPs to vanilla SEPs (Appendix B.3).
* Add a demonstration of how SEP predictions can improve the accuracy of models by refusing to answer for the most uncertain set of model generations (Appendix A, ‘Rejection-Accuracy Curve for SE’).
* Improve discussion of why SEPs generalize better than accuracy probes in Section 8.
* Improve exhibition of related work on probing in Section 2.

We further hope to address the individual questions and concerns that you have raised in our replies to your reviews below. With this rebuttal, we have also updated the paper draft based on your helpful feedback. For your convenience, we have highlighted all changes using a blue font color, and will undo this for the camera ready version. Again, thank you for your reviews, and we hope for active discussion with you to alleviate any remaining concerns over the next days!

---

### Meta-Review · Area_Chair_VVZZ · 2024-12-20

**Metareview:**

This work proposes Semantic Entropy Probes (SEPs), a method for uncertainty quantification in Large Language Models (LLMs) that approximates semantic entropy directly from hidden states, avoiding the computational overhead of sampling multiple generations. Experiments have demonstrated that SEPs retain high performance in detecting hallucinations, generalize well to out-of-distribution data, and significantly reduce the cost of semantic uncertainty quantification during inference. During the rebuttal phase, the authors provided additional experiments and explanations to address the reviewers' concerns. While many of these concerns have been successfully addressed, some remain unresolved. First, the reviewer still has significant concerns regarding the performance of accuracy probing, which outperforms the proposed Semantic Entropy Probe on in-distribution data. Although the authors presented experiments highlighting SEP’s advantages on out-of-distribution data, the reviewer remains concerned due to the lack of a solid explanation for why and how this occurs. Second, generating semantic entropy during training incurs substantial overhead due to the cost of providing semantic entropy ground truth. Therefore, we have decided not to accept this work in its current state.

**Additional Comments On Reviewer Discussion:**

We have carefully reviewed the submission, along with the discussion between the authors and reviewers during the rebuttal phase. We hope the authors will incorporate the reviewers' feedback to further improve this submission.

---

### Decision · Program_Chairs · 2025-01-22

Reject